# Flexible Attention-Based Multi-Policy Fusion for Efficient Deep Reinforcement Learning

**Zih-Yun Chiu**[1*]   **Yi-Lin Tuan**[2*]   **William Yang Wang**[2]   **Michael C. Yip**[1]
[1]University of California, San Diego   [2]University of California, Santa Barbara

## Abstract

Reinforcement learning (RL) agents have long sought to approach the efficiency of human learning. Humans are great observers who can learn by aggregating external knowledge from various sources, including observations from others' policies of attempting a task. Prior studies in RL have incorporated external knowledge policies to help agents improve sample efficiency. However, it remains non-trivial to perform arbitrary combinations and replacements of those policies, an essential feature for generalization and transferability. In this work, we present Knowledge-Grounded RL (KGRL), an RL paradigm fusing multiple knowledge policies and aiming for human-like efficiency and flexibility. We propose a new actor architecture for KGRL, Knowledge-Inclusive Attention Network (KIAN), which allows free knowledge rearrangement due to embedding-based attentive action prediction. KIAN also addresses entropy imbalance, a problem arising in maximum entropy KGRL that hinders an agent from efficiently exploring the environment, through a new design of policy distributions. The experimental results demonstrate that KIAN outperforms alternative methods incorporating external knowledge policies and achieves efficient and flexible learning. Our implementation is available at `https://github.com/Pascalson/KGRL.git`.

## 1   Introduction

Reinforcement learning (RL) has been effectively used in a variety of fields, including physics [7, 35] and robotics [15, 30]. This success can be attributed to RL's iterative process of interacting with the environment and learning a policy to get positive feedback. Despite being influenced by the learning process of infants [32], the RL process can require a large number of samples to solve a task [1], indicating that the learning efficiency of RL agents is still far behind that of humans.

*What learning capabilities do humans possess, yet RL agents still missing?* Studies in social learning [4] have demonstrated that humans often observe the behavior of others in diverse situations and utilize those strategies as *external knowledge* to accelerate their own exploration of solution-space. This type of learning is very flexible for humans since they can freely reuse and update the knowledge they already possess. The followings are the five properties (the last four have been mentioned in [14]) that summarize the efficiency and flexibility of human learning. **[Knowledge-Acquirable]**: Humans can develop their strategies by observing others. **[Sample-Efficient]**: Humans require fewer interactions with the environment to solve a task by learning from external knowledge. **[Generalizable]**: Humans can apply previously observed strategies, whether developed internally or provided externally, to unseen tasks. **[Compositional]**: Humans can combine strategies from multiple sources to form their knowledge set. **[Incremental]**: Humans do not need to relearn how to navigate the entire knowledge set from scratch when they remove outdated strategies or add new ones.

---

*indicates equal contribution. The corresponding emails are `zchiu@ucsd.edu` and `ytuan@cs.ucsb.edu`

37th Conference on Neural Information Processing Systems (NeurIPS 2023).

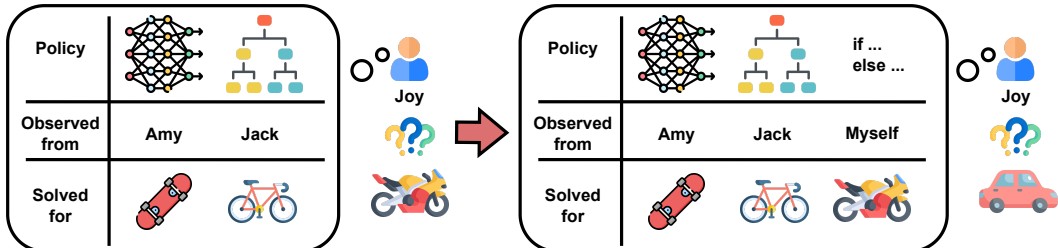

Figure 1: An illustration of knowledge-acquirable, compositional, and incremental properties in KGRL. Joy first learns to ride a motorcycle by observing Amy skateboarding and Jack biking. Then Joy learns to drive a car with the knowledge set expanded by Joy's developed strategy of motorcycling.

Possessing all five learning properties remains challenging for RL agents. Previous work has endowed an RL agent with the ability to learn from external knowledge (knowledge-acquirable) and mitigate sample inefficiency [21, 25, 27, 36], where the knowledge focused in this paper is state-action mappings (full definition in Section 3), including pre-collected demonstrations or policies. Among those methods, some have also allowed agents to combine policies in different forms to predict optimal actions (compositional) [25, 27]. However, these approaches may not be suitable for incremental learning, in which an agent learns a sequence of tasks using one expandable knowledge set. In such a case, whenever the knowledge set is updated by adding or replacing policies, prior methods, e.g., [27, 36], require relearning the entire multi-policy fusion process, even if the current task is similar to the previous one. This is because their designs of knowledge representations are intertwined with the knowledge-fusing mechanism, which restricts changing the number of policies in the knowledge set.

To this end, our goal is to enhance RL *grounded on external knowledge policies* with more flexibility. We first introduce *Knowledge-Grounded Reinforcement Learning (KGRL)*, an RL paradigm that seeks to find an optimal policy of a Markov Decision Process (MDP) given a set of external policies as illustrated in Figure 1. We then formally define the knowledge-acquirable, sample-efficient, generalizable, compositional, and incremental properties that a well-trained KGRL agent can possess.

We propose a simple yet effective actor model, *Knowledge-Inclusive Attention Network (KIAN)*, for KGRL. KIAN consists of three components: (1) an internal policy that learns a self-developed strategy, (2) embeddings that represent each policy, and (3) a query that performs *embedding-based attentive action prediction* to fuse the internal and external policies. The policy-embedding and query design in KIAN is crucial, as it enables the model to be incremental by unifying policy representations and separating them from the policy-fusing process. Consequently, updating or adding policies to KIAN has minimal effect on its architecture and does not require retraining the entire network. Additionally, KIAN addresses the problem of *entropy imbalance* in KGRL, where agents tend to choose only a few sub-optimal policies from the knowledge set. We provide mathematical evidence that entropy imbalance can prevent agents from exploring the environment with multiple policies. Then we introduce a new approach for modeling external-policy distributions to mitigate this issue.

Through experiments on grid navigation [5] and robotic manipulation [24] tasks, KIAN outperforms alternative methods incorporating external policies in terms of sample efficiency as well as the ability to do compositional and incremental learning. Furthermore, our analyses suggest that KIAN has better generalizability when applied to environments that are either simpler or more complex.

Our contributions are:

- We introduce KGRL, an RL paradigm studying how agents learn with external policies while being knowledge-acquirable, sample-efficient, generalizable, compositional, and incremental.

- We propose KIAN, an actor model for KGRL that fuses multiple knowledge policies with better flexibility and addresses entropy imbalance for more efficient exploration.

- We demonstrate in experiments that KIAN outperforms other methods incorporating external knowledge policies under different environmental setups.

## 2 Related Work

A popular line of research in RL is to improve sample efficiency with demonstrations (RL from demonstrations; RLfD). Demonstrations are examples of completing a task and are represented as state-action pairs. Previous work has leveraged demonstrations by introducing them into the policy-update steps of RL [8, 11, 21, 23, 28, 34]. For example, Nair et al. [21] adds a buffer of demonstrations to the RL framework and uses the data sampled from it to calculate a behavior-cloning loss. This loss is combined with the regular RL loss to make the policy simultaneously imitate demonstrations and maximize the expected return. RLfD methods necessitate an adequate supply of high-quality demonstrations to achieve sample-efficient learning, which can be time-consuming. In addition, they are low-level representations of a policy. Consequently, if an agent fails to extract a high-level strategy from these demonstrations, it will merely mimic the actions without acquiring a generalizable policy. In contrast, our proposed KIAN enables an agent to learn with external policies of arbitrary quality and fuse them by evaluating *the importance of each policy to the task*. Thus, the agent must understand the high-level strategies of each policy rather than only imitating its actions.

Another research direction in RL focuses on utilizing sub-optimal external policies instead of demonstrations to improve sample efficiency [25, 27, 36]. For instance, Zhang et al. [36] proposed Knowledge-Guided Policy Network (KoGuN) that learns a neural network policy from fuzzy-rule controllers. The neural network concatenates a state and all actions suggested by fuzzy-rule controllers as an input and outputs a refined action. While effective, this method puts restrictions on the representation of a policy to be a fuzzy logic network. On the other hand, Rajendran et al. [27] presented A2T (Attend, Adapt, and Transfer), an attentive deep architecture that fuses multiple policies and does not restrict the form of a policy. These policies can be non-primitive, and a learnable internal policy is included. In A2T, an attention network takes a state as an input and outputs the weights of all policies. The agent then samples an action from the fused distribution based on these weights. The methods KoGuN and A2T are most related to our work. Based on their success, KIAN further relaxes their requirement of retraining for incremental learning since both of them depend on the preset number of policies. Additionally, our approach mitigates the entropy imbalance issue, which can lead to inefficient exploration and was not addressed by KoGuN and A2T.

There exist other RL frameworks, such as hierarchical RL (HRL), that tackle tasks involving multiple policies. However, these frameworks are less closely related to our work compared to the previously mentioned methods. HRL approaches aim to decompose a complex task into a hierarchy of sub-tasks and learn a sub-policy for each sub-task [2, 6, 13, 16–18, 20, 25, 31, 33]. On the other hand, KGRL methods, including KoGuN, A2T, and KIAN, aim to address a task by observing a given set of external policies. These policies may offer partial solutions, be overly intricate, or even have limited relevance to the task at hand. Furthermore, HRL methods typically apply only one sub-policy to the environment at each time step based on the high-level policy, which determines the sub-task the agent is currently addressing. In contrast, KGRL seeks to simultaneously apply multiple policies within a single time step by fusing them together.

## 3 Problem Formulation

Our goal is to investigate how RL can be grounded on any given set of external knowledge policies to achieve knowledge-acquirable, sample-efficient, generalizable, compositional, and incremental properties. We refer to this RL paradigm as *Knowledge-Grounded Reinforcement Learning (KGRL)*.

A KGRL problem is a sequential decision-making problem that involves an environment, an agent, and a set of external policies. It can be mathematically formulated as a Knowledge-Grounded Markov Decision Process (KGMDP), which is defined by a tuple $\mathcal{M}_k = (\mathcal{S}, \mathcal{A}, \mathcal{T}, R, \rho, \gamma, \mathcal{G})$, where $\mathcal{S}$ is the state space, $\mathcal{A}$ is the action space, $\mathcal{T} : \mathcal{S} \times \mathcal{A} \times \mathcal{S} \to \mathbb{R}$ is the transition probability distribution, $R$ is the reward function, $\rho$ is the initial state distribution, $\gamma$ is the discount factor, and $\mathcal{G}$ is the set of external knowledge policies. An external knowledge set $\mathcal{G}$ contains $n$ knowledge policies, $\mathcal{G} = \{\pi_{g_1}, \ldots, \pi_{g_n}\}$. Each knowledge policy is a function that maps from the state space to the action space, $\pi_{g_j}(\cdot|\cdot) : \mathcal{S} \to \mathcal{A}, \forall j = 1, \ldots, n$. A knowledge mapping is not necessarily designed for the original Markov Decision Process (MDP), which is defined by the tuple $\mathcal{M} = (\mathcal{S}, \mathcal{A}, \mathcal{T}, \mathcal{R}, \rho, \gamma)$. Therefore, applying $\pi_{g_j}$ to $\mathcal{M}$ may result in a poor expected return.

The goal of KGRL is to find an optimal policy $\pi^*(\cdot|\cdot;\mathcal{G}) : \mathcal{S} \to \mathcal{A}$ that maximizes the expected return: $\mathbb{E}_{\mathbf{s}_0 \sim \rho, \mathcal{T}, \pi^*}[\sum_{t=0}^{T} \gamma^t R_t]$. Note that $\mathcal{M}_k$ and $\mathcal{M}$ share the same optimal value function, $V^*(\mathbf{s}) = \max_{\pi \in \Pi} \mathbb{E}_{\mathcal{T}, \pi}[\sum_{k=0}^{\infty} \gamma^k R_{t+k+1}|\mathbf{s}_t = \mathbf{s}]$, if they are provided with the same policy class $\Pi$.

A well-trained KGRL agent can possess the following properties: knowledge-acquirable, sample-efficient, generalizable, compositional, and incremental. Here we formally define these properties.

**Definition 3.1** (Knowledge-Acquirable). An agent can acquire knowledge internally instead of only following $\mathcal{G}$. We refer to this internal knowledge as *an inner policy* and denote it as $\pi_{in}(\cdot|\cdot) : \mathcal{S} \to \mathcal{A}$.

**Definition 3.2** (Sample-Efficient). An agent requires fewer samples to solve for $\mathcal{M}_k$ than for $\mathcal{M}$.

**Definition 3.3** (Generalizable). A learned policy $\pi(\cdot|\cdot;\mathcal{G})$ can solve similar but different tasks.

**Definition 3.4** (Compositional). Assume that other agents have solved for $m$ KGMDPs, $\mathcal{M}_k^1, \ldots, \mathcal{M}_k^m$, with external knowledge sets, $\mathcal{G}^1, \ldots, \mathcal{G}^m$, and inner policies, $\pi_{in}^1, \ldots, \pi_{in}^m$. An agent is *compositional* if it can learn to solve a KGMDP $\mathcal{M}_k^*$ with the external knowledge set $\mathcal{G}^* \subseteq \bigcup_{i=1}^{m} \mathcal{G}^i \cup \{\pi_{in}^1, \ldots, \pi_{in}^m\}$.

**Definition 3.5** (Incremental). An agent is *incremental* if it has the following two abilities: (1) Given a KGMDP $\mathcal{M}_k$ for the agent to solve within $T$ timesteps. The agent can learn to solve $\mathcal{M}_k$ with the external knowledge sets, $\mathcal{G}_1, \ldots, \mathcal{G}_T$, where $\mathcal{G}_t, t \in \{1, \ldots, T\}$, is the knowledge set at time step $t$, and $\mathcal{G}_t$ can be different from one another. (2) Given a sequence of KGMDPs $\mathcal{M}_k^1, \ldots, \mathcal{M}_k^m$, the agent can solve them with external knowledge sets, $\mathcal{G}^1, \ldots, \mathcal{G}^m$, where $\mathcal{G}^i, i \in \{1, \ldots, m\}$, is the knowledge set for task $i$, and $\mathcal{G}^i$ can be different from one another.

## 4 Knowledge-Inclusive Attention Network

We propose Knowledge-Inclusive Attention Network (KIAN) as an actor for KGRL. KIAN can be end-to-end trained with various RL algorithms. Illustrated in Figure 2, KIAN comprises three components: an inner actor, knowledge keys, and a query. In this section, we first describe the architecture of KIAN and its action-prediction operation. Then we introduce entropy imbalance, a problem that emerges in maximum entropy KGRL, and propose modified policy distributions for KIAN to alleviate this issue.

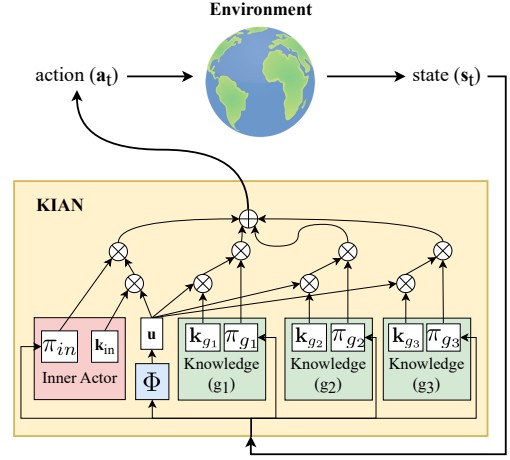

Figure 2: The model architecture of KIAN.

### 4.1 Model Architecture

**Inner Actor.** The inner actor serves the same purpose as an actor in regular RL, representing the *inner knowledge* learned by the agent through interactions with the environment. In KIAN, the inner actor, denoted as $\pi_{in}(\cdot|\cdot;\boldsymbol{\theta}) : \mathcal{S} \to \mathcal{A}$, is a learnable function approximator with parameter $\boldsymbol{\theta}$. The presence of the inner actor in KIAN is crucial for the agent to be capable of acquiring knowledge, as it allows the agent to develop its own strategies. Therefore, even if the external knowledge policies in $\mathcal{G}$ are unable to solve a particular task, the agent can still discover an optimal solution.

**Knowledge Keys.** In KIAN, we introduce a learnable embedding vector for each knowledge policy, including $\pi_{in}$ and $\pi_{g_1}, \ldots, \pi_{g_n}$, in order to create a unified representation space for all knowledge policies. Specifically, for each knowledge mapping $\pi_{in}$ or $\pi_{g_j} \in \mathcal{G}$, we assign a learnable $d_k$-dimensional vector as its key (embedding): $\mathbf{k}_{in} \in \mathbb{R}^{d_k}$ or $\mathbf{k}_{g_j} \in \mathbb{R}^{d_k} \; \forall j \in \{1, \ldots, n\}$. It is important to note that these knowledge keys, $\mathbf{k}_e$, represents the entire knowledge mapping $\pi_e, \forall e \in \{in, g_1, \ldots, g_n\}$. Thus, $\mathbf{k}_e$ is independent of specific states or actions. These knowledge keys and the query will perform an attention operation to determine how an agent integrates all policies.

Our knowledge-key design is essential for an agent to be compositional and incremental. By unifying the representation of policies through knowledge keys, we remove restrictions on the form of a

knowledge mapping. It can be any form, such as a lookup table of state-action pairs (demonstrations) [21], if-else-based programs, fuzzy logics [36], or neural networks [25, 27]. In addition, the knowledge keys are not ordered, so $\pi_{g_1}, \ldots, \pi_{g_n}$ in $\mathcal{G}$ and their corresponding $\mathbf{k}_{g_1}, \ldots, \mathbf{k}_{g_n}$ can be freely rearranged. Finally, since a knowledge policy is encoded as a key *independent of other knowledge keys* in a joint embedding space, replacing a policy in $\mathcal{G}$ means replacing a knowledge key in the embedding space. This replacement requires no changes in the other part of KIAN's architecture. Therefore, an agent can update $\mathcal{G}$ anytime without relearning a significant part of KIAN.

**Query.** The last component in KIAN, *the query*, is a function approximator that generates $d_k$-dimensional vectors for knowledge-policy fusion. The query is learnable with parameter $\phi$ and is state-dependent, so we denote it as $\Phi(\cdot; \phi) : \mathcal{S} \to \mathbb{R}^{d_k}$. Given a state $\mathbf{s}_t \in \mathcal{S}$, the query outputs a $d_k$-dimensional vector $\mathbf{u}_t = \Phi(\mathbf{s}_t; \phi) \in \mathbb{R}^{d_k}$, which will be used to perform *an attention operation* with all knowledge keys. This operation determines *the weights* of policies when fusing them.

## 4.2 Embedding-Based Attentive Action Prediction

The way to predict an action with KIAN and a set of external knowledge policies, $\mathcal{G}$, is by three steps: (1) calculating a weight for each knowledge policy using an embedding-based attention operation, (2) fusing knowledge policies with these weights, and (3) sampling an action from the fused policy.

**Embedding-Based Attention Operation.** Given a state $\mathbf{s}_t \in \mathcal{S}$, KIAN predicts a weight for each knowledge policy as *how likely this policy will suggest a good action*. These weights can be computed by the dot product between the query and knowledge keys as:

$$\begin{aligned}
w_{t,in} &= \Phi(\mathbf{s}_t; \phi) \cdot \mathbf{k}_{in}/c_{t,in} \in \mathbb{R}, \\
w_{t,g_j} &= \Phi(\mathbf{s}_t; \phi) \cdot \mathbf{k}_{g_j}/c_{t,g_j} \in \mathbb{R}, \quad \forall j \in \{1, \ldots, n\}.
\end{aligned} \tag{1}$$

$$[\hat{w}_{t,in}, \hat{w}_{t,g_1}, \ldots, \hat{w}_{t,g_n}]^\top = \texttt{softmax}([w_{t,in}, w_{t,g_1}, \ldots, w_{t,g_n}]^\top). \tag{2}$$

where $c_{t,in} \in \mathbb{R}$ and $c_{t,g_j} \in \mathbb{R}$ are normalization factors, for example, if $c_{t,g_j} = \|\Phi(\mathbf{s}_t; \phi)\|_2 \|\mathbf{k}_{g_j}\|_2$, then $w_{t,g_j}$ turns out to be the cosine similarity between $\Phi(\mathbf{s}_t; \phi)$ and $\mathbf{k}_{g_j}$. We refer to this operation as *an embedding-based attention operation* since the query evaluates each knowledge key (embedding) by equation (1) to determine how much attention an agent should pay to the corresponding knowledge policy. If $w_{t,in}$ is larger than $w_{t,g_j}$, the agent relies more on its self-learned knowledge policy $\pi_{in}$; otherwise, the agent depends more on the action suggested by the knowledge policy $\pi_{g_j}$. Note that the computation of one weight is independent of other knowledge keys, so changing the number of knowledge policies will not affect the relation among all remaining knowledge keys.

**Action Prediction for A Discrete Action Space.** An MDP (or KGMDP) with a discrete action space usually involves choosing from $d_a \in \mathbb{N}$ different actions, so each knowledge policy maps from a state to *a $d_a$-dimensional probability simplex*, $\pi_{in} : \mathcal{S} \to \Delta^{d_a}, \pi_{g_j} : \mathcal{S} \to \Delta^{d_a} \forall j = 1, \ldots, n$. When choosing an action given a state $\mathbf{s}_t \in \mathcal{S}$, KIAN first predicts $\pi(\cdot|\mathbf{s}_t) \in \Delta^{d_a} \subseteq \mathbb{R}^{d_a}$ with the weights, $\hat{w}_{in}, \hat{w}_{g_1}, \ldots, \hat{w}_{g_n}$:

$$\pi(\cdot|\mathbf{s}_t) = \hat{w}_{in}\pi_{in}(\cdot|\mathbf{s}_t) + \Sigma_{j=1}^n \hat{w}_{g_j} \pi_{g_j}(\cdot|\mathbf{s}_t), \tag{3}$$

The final action is sampled as $a_t \sim \pi(\cdot|\mathbf{s}_t)$, where the $i$-th element of $\pi(\cdot|\mathbf{s}_t)$ represents the probability of sampling the $i$-th action.

**Action Prediction for A Continuous Action Space.** Each knowledge policy for a continuous action space is a probability distribution that suggests a $d_a$-dimensional action for an agent to apply to the task. As prior work [25], we model each knowledge policy as a multivariate normal distribution, $\pi_{in}(\cdot|\mathbf{s}_t) = \mathcal{N}(\boldsymbol{\mu}_{t,in}, \boldsymbol{\sigma}_{t,in}^2), \pi_{g_j}(\cdot|\mathbf{s}_t) = \mathcal{N}(\boldsymbol{\mu}_{t,g_j}, \boldsymbol{\sigma}_{t,g_j}^2) \forall j \in \{1, \ldots, n\}$, where $\boldsymbol{\mu}_{t,in} \in \mathbb{R}^{d_a}$ and $\boldsymbol{\mu}_{t,g_j} \in \mathbb{R}^{d_a}$ are the means, and $\boldsymbol{\sigma}_{t,in}^2 \in \mathbb{R}_{\geq 0}^{d_a}$ and $\boldsymbol{\sigma}_{t,g_j}^2 \in \mathbb{R}_{\geq 0}^{d_a}$ are the diagonals of the covariance matrices. Note that we assume each random variable in an action is independent of one another.

A continuous policy fused as equation (3) becomes a mixture of normal distributions. To sample an action from this mixture of distributions without losing the important information provided by each distribution, we choose only one knowledge policy according to the weights and sample an action from it. We first sample an element from the set

$\{in, g_1, \ldots, g_n\}$ according to the weights, $\{\hat{w}_{t,in}, \hat{w}_{t,g_1}, \ldots, \hat{w}_{t,g_n}\}$, using Gumbel softmax [12]: $e \sim \texttt{gumbel\_softmax}([\hat{w}_{t,in}, \hat{w}_{t,g_1}, \ldots, \hat{w}_{t,g_n}]^\top)$, in order to make KIAN differentiable everywhere. Then given a state $\mathbf{s}_t \in \mathcal{S}$, an action is sampled from the knowledge policy, $\mathbf{a}_t \sim \pi_e(\cdot|\mathbf{s}_t)$, using the reparameterization trick.

However, fusing multiple policies as equation (3) will make an agent biased toward a small set of knowledge policies when exploring the environment in the context of maximum entropy KGRL.

### 4.3 Exploration in KGRL

Maximizing entropy is a commonly used approach to encourage exploration in RL [9, 10, 37]. However, in maximum entropy KGRL, when the entropy of policy distributions are different from one another, it leads to the problem of *entropy imbalance*. Entropy imbalance is a phenomenon in which an agent consistently selects only a single or a small set of knowledge policies. We show this in math by first revisiting the formulation of maximum entropy RL. In maximum entropy RL, an entropy term is added to the standard RL objective as $\pi^* = \arg\max_\pi \sum_t \mathbb{E}_{(\mathbf{s}_t, \mathbf{a}_t \sim \pi)} [R(\mathbf{s}_t, \mathbf{a}_t) + \alpha H(\pi(\cdot|\mathbf{s}_t))]$ [9, 10], where $\alpha \in \mathbb{R}$ is a hyperparameter, and $H(\cdot)$ represents the entropy of a distribution. By maximizing $\alpha H(\pi(\cdot|\mathbf{s}_t))$, the policy becomes more uniform since the entropy of a probability distribution is maximized when it is a uniform distribution [19]. With this in mind, we show that in maximum entropy KGRL, some of the weights in $\{\hat{w}_{t,in}, \hat{w}_{t,g_1}, \ldots, \hat{w}_{t,g_n}\}$ might always be larger than others. We provide the proofs of all propositions in Appendix A.

**Proposition 4.1** (Entropy imbalance in discrete decision-making). *Assume that a $d_a$-dimensional probability simplex $\pi \in \Delta^{d_a}$ is fused by $\{\pi_1, \ldots, \pi_m\}$ and $\{\hat{w}_1, \ldots, \hat{w}_m\}$ following equation (3), where $\pi_j \in \Delta^{d_a}, \hat{w}_j \geq 0 \; \forall j \in \{1, \ldots, m\}$ and $\sum_{j=1}^m \hat{w}_j = 1$. If the entropy of $\pi$ is maximized and $\|\pi_1\|_\infty \ll \|\pi_2\|_\infty, \|\pi_1\|_\infty \ll \|\pi_3\|_\infty, \ldots, \|\pi_1\|_\infty \ll \|\pi_m\|_\infty$, then $\hat{w}_1 \to 1$.*

We show in Proposition A.1 that if $\pi_1$ is *more uniform* than $\pi_j$, then $\|\pi_1\|_\infty < \|\pi_j\|_\infty$.

**Proposition 4.2** (Entropy imbalance in continuous control). *Assume a one-dimensional policy distribution $\pi$ is fused by*

$$\pi = \hat{w}_1 \pi_1 + \hat{w}_2 \pi_2, \text{ where } \pi_j = \mathcal{N}(\mu_j, \sigma_j^2), \hat{w}_j \geq 0 \; \forall j \in \{1, 2\}, \text{ and } \hat{w}_1 + \hat{w}_2 = 1. \quad (4)$$

*If the variance of $\pi$ is maximized, and $\sigma_1^2 \gg \sigma_2^2$ and $\sigma_1^2 \gg (\mu_1 - \mu_2)^2$, then $\hat{w}_1 \to 1$.*

We can also infer from Proposition 4.2 that the variance of $\pi$ defined in equation (4) depends on the distance between $\mu_1$ and $\mu_2$, which leads to Proposition 4.3.

**Proposition 4.3** (Distribution separation in continuous control). *Assume a one-dimensional policy distribution $\pi$ is fused by equation (4). If $\hat{w}_1, \hat{w}_2, \sigma_1^2$, and $\sigma_2^2$ are fixed, then maximizing the variance of $\pi$ will increase the distance between $\mu_1$ and $\mu_2$.*

Proposition 4.1, 4.2, and 4.3 indicate that in maximum entropy KGRL, (1) the agent will pay more attention to the policy with large entropy, and (2) in continuous control, an agent with a learnable internal policy will rely on this policy and separate it as far away as possible from other policies. The consistently imbalanced attention prevents the agent from exploring the environment with other policies that might provide helpful suggestions to solve the task. Furthermore, in continuous control, the distribution separation can make $\pi$ perform even worse than learning without any external knowledge. The reason is that external policies, although possibly being sub-optimal for the task, might be more efficient in approaching the goal, and moving away from those policies means being less efficient when exploring the environment.

### 4.4 Modified Policy Distributions

Proposition 4.1 and 4.2 show that fusing multiple policies with equation (3) can make a KGRL agent rely on a learnable internal policy for exploration. However, the uniformity of the internal policy is often desired since it encourages exploration in the state-action space that is not covered by external policies. Therefore, we keep the internal policy unchanged and propose methods to modify external policy distributions in KIAN to resolve the entropy imbalance issue. We provide the detailed learning algorithm of KGRL with KIAN in Appendix A.6.

**Discrete Policy Distribution.** We modify a fusion of discrete policy distributions in equation (3) as

$$\pi(\cdot|\mathbf{s}_t) = \hat{w}_{t,in}\pi_{in}(\cdot|\mathbf{s}_t) + \Sigma_{j=1}^n \hat{w}_{t,g_j}\texttt{softmax}(\beta_{t,g_j}\pi_{g_j}(\cdot|\mathbf{s}_t)), \tag{5}$$

$$w_{t,in} = \frac{\Phi(\mathbf{s}_t) \cdot \mathbf{k}_{in}}{\|\Phi(\mathbf{s}_t)\|_2\|\mathbf{k}_{in}\|_2}, w_{t,g_j} = \frac{\Phi(\mathbf{s}_t) \cdot \mathbf{k}_{g_j}}{\|\Phi(\mathbf{s}_t)\|_2\|\mathbf{k}_{g_j}\|_2}, \tag{6}$$

$$\beta_{t,g_j} = \|\Phi(\mathbf{s}_t)\|_2\|\mathbf{k}_{g_j}\|_2 \quad \forall j \in \{1,\ldots,n\}, \tag{7}$$

where $\beta_{t,g_j} \in \mathbb{R}$ is a state-and-knowledge dependent variable that scales $\pi_{g_j}(\cdot|\mathbf{s}_t)$ to change its uniformity after passing through $\texttt{softmax}$. If the value of $\beta_{t,g_j}$ decreases, the uniformity, i.e., the entropy, of $\texttt{softmax}(\beta_{t,g_j}\pi_{g_j}(\cdot|\mathbf{s}_t))$ increases. By introducing $\beta_{t,g_j}$, the entropy of knowledge policies becomes adjustable, resulting in reduced bias towards the internal policy during exploration.

**Continuous Action Probability.** We modify *the probability of sampling* $\mathbf{a}_t \in \mathbb{R}^{d_a}$ from a continuous $\pi(\cdot|\mathbf{s}_t)$ in equation (3) as

$$\pi(\mathbf{a}_t|\mathbf{s}_t) = \hat{w}_{in}\pi_{in}(\mathbf{a}_{t,in}|\mathbf{s}_t) + \Sigma_{j=1}^n \hat{w}_{g_j}\pi_{g_j}(\boldsymbol{\mu}_{t,g_j}|\mathbf{s}_t), \tag{8}$$

where $\mathbf{a}_{t,in} \sim \pi_{in}(\cdot|\mathbf{s}_t)$ and $\boldsymbol{\mu}_{t,g_j} \in \mathbb{R}^{d_a}$ is the mean of $\pi_{g_j}(\cdot|\mathbf{s}_t)$. We show in the next proposition that equation (8) is an approximation of

$$\pi(\mathbf{a}_t|\mathbf{s}_t) = \hat{w}_{in}\pi_{in}(\mathbf{a}_t|\mathbf{s}_t) + \Sigma_{j=1}^n \hat{w}_{g_j}\pi_{g_j}(\mathbf{a}_t|\mathbf{s}_t), \tag{9}$$

which is the exact probability of sampling $\mathbf{a}_t \in \mathbb{R}^{d_a}$ from a continuous $\pi(\cdot|\mathbf{s}_t)$ in equation (3).

**Proposition 4.4** (Approximation of a mixture of normal distributions)**.** *If the following three inequalities hold for* $\mu_{t,in}, \mu_{t,g_1}, \ldots, \mu_{t,g_n}$, *and* $a_{t,in}$: $\|\mu_{t,in} - \mu_{t,g_j}\|_2 < min\{\gamma_{t,in}, \gamma_{t,g_j}\}$, $\|a_{t,in} - \mu_{t,in}\|_2 < min\{\gamma_{t,in}, \gamma_{t,g_j}\}$, *and* $\|a_{t,in} - \mu_{t,g_j}\|_2 < \gamma_{t,g_j}$, $\forall j \in \{1,\ldots,n\}$, *where* $\gamma_{t,in} = 1/(2\pi_{in}(\mu_{t,in}|\mathbf{s}_t))$ *and* $\gamma_{t,g_j} = 1/(2\pi_{g_j}(\mu_{t,g_j}|\mathbf{s}_t))$, *then equation (9) for a real-valued action* $a_t$ *sampled from KIAN can be approximated by*

$$\hat{w}_{t,in}\mathcal{U}(a_t; \mu_{t,in} - \gamma_{t,in}, \mu_{t,in} + \gamma_{t,in}) + \sum_{j=1}^n \hat{w}_{t,g_j}\mathcal{U}(a_t; \mu_{t,in} - \gamma_{t,g_j}, \mu_{t,in} + \gamma_{t,g_j}), \tag{10}$$

$$where \quad \mathcal{U}(\cdot; a, b) = 1/(b-a). \tag{11}$$

*In addition, equation (8) is a lower bound of equation (10).*

With equation (8), we can show that maximizing the variance of $\pi(\cdot|\mathbf{s}_t)$ will not separate the policy distributions. Hence, an agent can refer to external policies for efficient exploration and learn its own refined strategy based on them.

**Proposition 4.5** (Maximized variance's independence of the distance between means)**.** *Assume a one-dimensional policy* $\pi$ *is fused by equation (4). If* $\pi(a|\mathbf{s})$ *is approximated as equation (8), and the three inequalities in Proposition 4.4 are satisfied, then maximizing the variance of* $\pi(\cdot|\mathbf{s})$ *will not affect the distance between* $\mu_1$ *and* $\mu_2$.

## 5 Experiments

We evaluate KIAN on two sets of environments with discrete and continuous action spaces: Mini-Grid [5] and OpenAI-Robotics [24]. Through experiments, we answer the following four questions: **[Sample Efficiency]** Does KIAN require fewer training samples to solve a task than other external-policy-inclusive methods? **[Generalizability]** Can KIAN trained on one task be directly used to solve another task? **[Compositional and Incremental Learning]** Can KIAN combine previously learned knowledge keys and inner policies to learn a new task? After adding more external policies to $\mathcal{G}$, can most of the components from a trained KIAN be reused for learning?

For comparison, we implement the following five methods as our baselines: behavior cloning (BC) [3], RL [10, 29], RL+BC [21], KoGuN [36], and A2T [27]. KoGuN and A2T are modified to be compositional and applicable in both discrete and continuous action spaces. Moreover, all methods (BC, RL+BC, KoGuN, A2T, and KIAN) are equipped with the same initial external knowledge set, $\mathcal{G}^{init}$, for each task. This knowledge set comprises sub-optimal if-else-based programs that cannot complete a task themselves, e.g., $\texttt{pickup\_a\_key}$ or $\texttt{move\_forward\_to\_the\_goal}$. $\mathcal{G}^{init}$ will be expanded with learned policies in compositional- and incremental-learning experiments. We provide the experimental details in Appendix B.

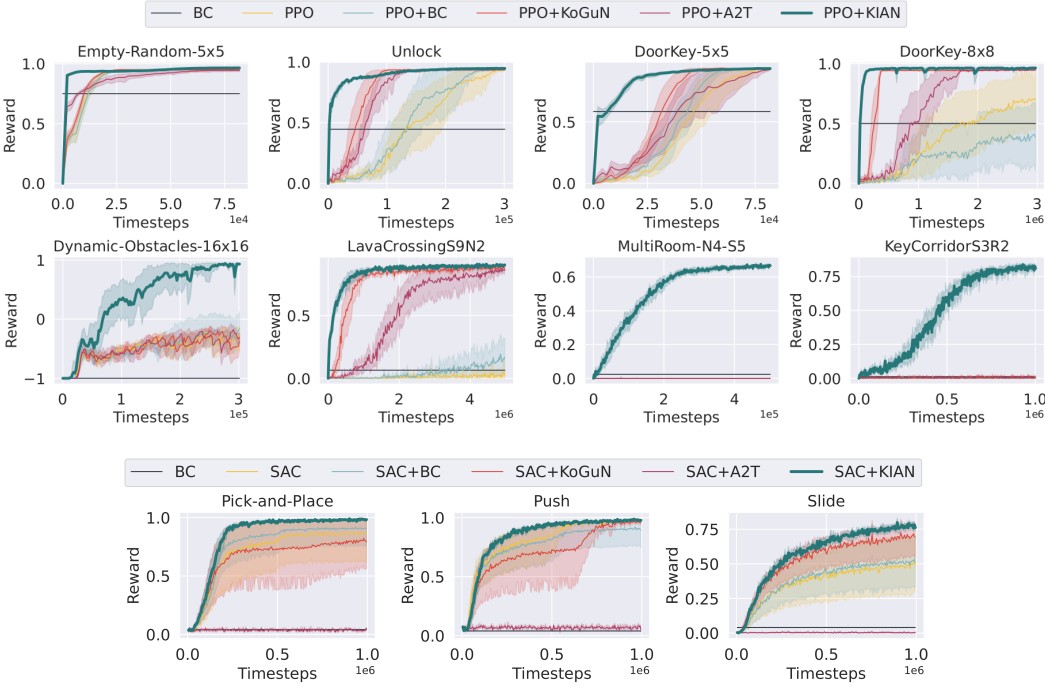

Figure 3: The learning curves of sample efficiency experiments in MiniGrid (top 2 rows) and OpenAI-Robotics (last row) environments. Given a knowledge set that cannot complete a task (as shown by BC), KIAN exhibits better sample efficiency across all tasks. These results underline the effectiveness of KIAN in leveraging external policies to mitigate the need for extensive training samples.

| Train in | Empty-Random-5x5 | | | DoorKey-5x5 | | Push | | Slide | | Pick-and-Place | |
| Test in | 6x6 | 8x8 | 16x16 | 8x8 | 16x16 | 5x | 10x | 5x | 10x | 5x | 10x |
|---|---|---|---|---|---|---|---|---|---|---|---|
| RL [10, 29] | 0.88 | 0.71 | 0.45 | 0.29 | 0.08 | 0.87 | 0.52 | 0.45 | 0.17 | 0.34 | 0.27 |
| RL+BC [21] | 0.87 | 0.60 | 0.24 | 0.40 | 0.09 | 0.89 | 0.60 | 0.44 | 0.16 | 0.34 | 0.30 |
| KoGuN [36] | 0.94 | 0.83 | 0.53 | **0.77** | 0.35 | 0.63 | 0.43 | **0.55** | **0.18** | 0.32 | 0.24 |
| A2T [27] | 0.92 | 0.78 | 0.51 | 0.53 | 0.11 | 0.03 | 0.05 | 0.00 | 0.01 | 0.01 | 0.06 |
| KIAN (ours) | **0.96** | **0.91** | **0.93** | 0.76 | **0.42** | **0.93** | **0.70** | 0.42 | 0.15 | **0.92** | **0.72** |

Table 1: (Zero-Shot S2C Experiments) The left five columns show the generalizability results of an agent trained in a 5x5 environment and tested in environments of varying sizes. The right six columns show the results of an agent trained with a 1x goal range and tested with different goal ranges. Transferring policies from a simple task to a more complex one is a challenging setup in generalizability experiments. The results highlight the superior performance of KIAN in such setup.

## 5.1 Sample Efficiency and Generalizability

We study the sample efficiency of baselines and KIAN under *the intra-task setup*, where an agent learns a single task with the external knowledge set $\mathcal{G}^{init}$ fixed. Figure 3 plots the learning curves in different environments. All experiments in these figures are run with ten random seeds, and each error band is a 95% confidence interval. The results of BC show that the external knowledge policies are sub-optimal for all environments. Given sub-optimal external knowledge, only KIAN shows success in all environments. In general, improvement of KIAN over baselines is more apparent when the task is more complex, e.g., Empty < Unlock < DoorKey and Push < Pick-and-Place. Moreover, KIAN is more stable than baselines in most environments. Note that in continuous-control tasks (Push, Slide, and Pick-and-Place), A2T barely succeeds since it does not consider the entropy imbalance issue introduced in Proposition 4.2 and 4.3. These results suggest that KIAN can more efficiently explore the environment with external knowledge policies and fuse multiple policies to solve a task.

Next, we evaluate the generalizability of all methods under *simple-to-complex (S2C)* and *complex-to-simple (C2S)* setups, where the former trains a policy in a simple task and test it in a complex one,

| Train in
Test in | DoorKey-5x5
Empty-Random | DoorKey-8x8
Unlock | DoorKey5x5 | Pick-and-Place
Reach | Push | Push
Reach | Slide
Push |
|---|---|---|---|---|---|---|---|
| RL [10, 29] | 0.83 | 0.92 | 0.93 | 0.80 | **0.31** | 0.16 | 0.09 |
| RL+BC [21] | 0.85 | 0.87 | 0.93 | 0.80 | **0.31** | 0.16 | 0.09 |
| KoGuN [36] | 0.90 | 0.91 | 0.93 | 0.45 | 0.05 | 0.20 | 0.07 |
| A2T [27] | 0.84 | 0.92 | 0.93 | 0.01 | 0.05 | 0.20 | 0.05 |
| KIAN (ours) | **0.91** | **0.94** | **0.95** | **1.00** | 0.30 | **0.24** | **0.13** |

Table 2: (Zero-Shot C2S Experiments) In general, KIAN outperforms other methods when transferring policies across different tasks. Note that although distinguishing the levels of difficulty between Push, Slide, and Pick-and-Place is not straightforward, KIAN still achieves better performance.

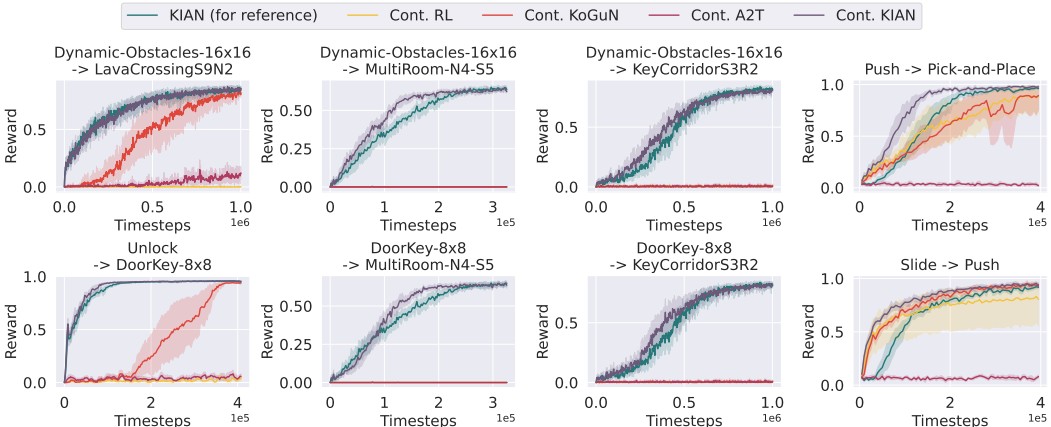

Figure 4: The learning curves of composition and incremental experiments in MiniGrid (left 3 columns) and OpenAI-Robotics (right column) environments. KIAN requires fewer samples to learn two tasks sequentially than separately and outperforms other approaches in incremental learning.

and the latter goes the opposite way. All generalizability experiments are run with the same policies as in Section 5.1. Table 1 and 2 show that KIAN outperforms other baselines in most experiments, and its results have a smaller variance (see Table 3 to 5 in Appendix E). These results demonstrate that KIAN's flexibility in incorporating external policies improves generalizability.

## 5.2 Compositional and Incremental Learning

In the final experiments, we test different methods in the compositional and incremental learning setting. We modify RL, KoGuN, and A2T to fit into this setting; details can be found in Appendix C. The experiments follow *the inter-task setup*: (1) We randomly select a pair of tasks $(\mathcal{M}_k^1, \mathcal{M}_k^2)$. (2) An agent learns a policy to solve $\mathcal{M}_k^1$ with $\mathcal{G}^{init}$ fixed, as done in Section 5.1. (3) The learned (internal) policy, $\pi_{in}^1$, is added to the external knowledge set, $\mathcal{G} = \mathcal{G}^{init} \cup \{\pi_{in}^1\}$. (4) The same agent learns a policy to solve $\mathcal{M}_k^2$ with $\mathcal{G}$. Each experiment is run with ten random seeds.

The learning curves in Figure 4 demonstrate that given the same updated $\mathcal{G}$, KIAN requires fewer samples to solve $\mathcal{M}_k^2$ than RL, KoGuN, and A2T in all experiments. Our knowledge-key and query design disentangles policy representations from the action-prediction operation, so the agent is more optimized in incremental learning. Unlike our disentangled design, prior methods use a single function approximator to directly predict an action (KoGuN) or the weight of each policy (A2T) given a state. These methods make the action-prediction operation depend on the number of knowledge policies, so changing the size of $\mathcal{G}$ requires significant retraining of the entire function approximator.

Figure 4 also shows that KIAN solves $\mathcal{M}_k^2$ more efficiently with $\mathcal{G}$ than $\mathcal{G}^{init}$ in most experiments. This improvement can be attributed to KIAN reusing the knowledge keys and query, which allows an agent to know which policies to fuse under different scenarios. Note that $\mathcal{G}$ can be further expanded with the internal policy learned in $\mathcal{M}_k^2$ and be used to solve another task $\mathcal{M}_k^3$.

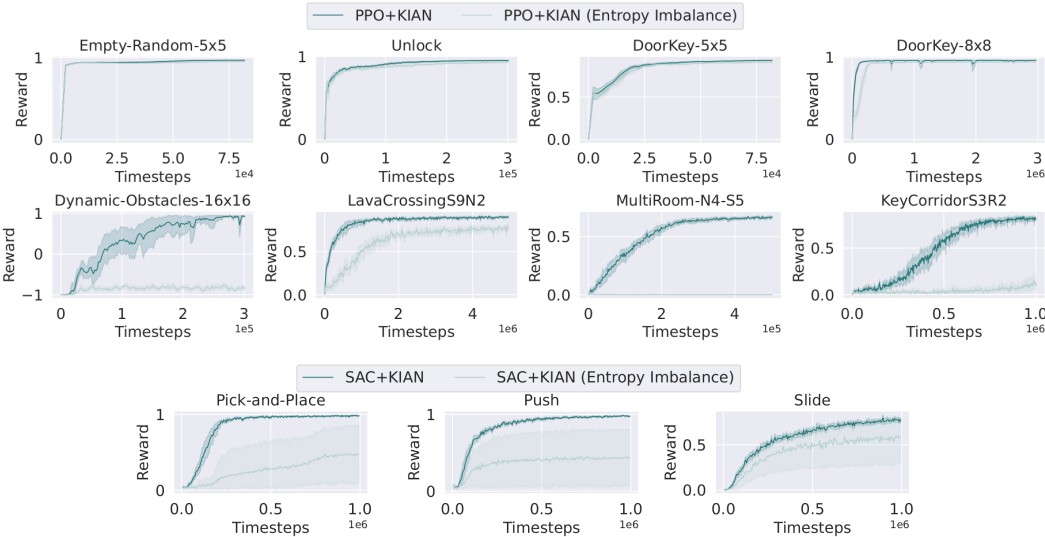

Figure 5: The learning curves of KIAN with and without addressing entropy imbalance as described in Section 4.4. The results indicate the adverse impact of entropy imbalance on KIAN's performance within the context of maximum entropy KGRL. In addition, our proposed modifications to external policy distributions are shown to be highly effective in alleviating this issue.

## 5.3 Analysis of Entropy Imbalance in Maximum Entropy KGRL

In our ablation study, we investigate (1) the impact of entropy imbalance on the performance of maximum entropy KGRL and (2) whether the proposed modifications to external policy distributions in Section 4.4 can alleviate the issue.

Figure 5 shows the learning curves comparing KIAN's performance with and without addressing the entropy-imbalance issue. The results demonstrate that when not addressing the issue using equation (5) or (8), KIAN fails to fully capitalize on the guidance offered by external policies. We also draw two noteworthy conclusions from the figure: (1) For discrete decision-making tasks, the detrimental impact of entropy imbalance becomes more evident as task complexity increases. (2) For continuous-control tasks, entropy imbalance can degrade KIAN's performance and make it perform worse than pure RL without external policies, as shown by the results of FetchPickAndPlace and FetchPush. This phenomenon can be attributed to Proposition 4.3. In contrast, by adjusting KIAN's external policy distributions using equation (5) or (8), a KGRL agent can efficiently harness external policies to solve a given task.

## 6 Conclusion and Discussion

This work introduces KGRL, an RL paradigm aiming to enhance efficient and flexible learning by harnessing external policies. We propose KIAN as an actor model for KGRL, which predicts an action by fusing multiple policies with an embedding-based attention operation. Furthermore, we propose modifications to KIAN's policy distributions to address entropy imbalance, which hinders efficient exploration with external policies in maximum entropy KGRL. Our experimental findings demonstrate that KIAN outperforms alternative methods incorporating external policies regarding sample efficiency, generalizability, and compositional and incremental learning.

However, it is essential to acknowledge a limitation not addressed in this work. The efficiency of KIAN, as well as other existing KGRL methods, may decrease when dealing with a large external knowledge set containing irrelevant policies. This issue is examined and discussed in Appendix F. Efficiently handling extensive sets of external policies is left for future research.

Our research represents an initial step towards the overarching goal of KGRL: learning a knowledge set with a diverse range of policies. These knowledge policies can be shared across various environments and continuously expanded, allowing artificial agents to flexibly query and learn from them. We provide detailed discussions on the broader impact of this work and outline potential directions of future research in Appendix D.

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

# A Proofs and Learning Algorithms of KIAN

## A.1 Proof of Proposition 4.1

**Proposition 4.1** (Entropy imbalance in discrete decision-making). *Assume that a $d_a$-dimensional probability simplex $\pi \in \Delta^{d_a}$ is fused by $\{\pi_1, \dots, \pi_m\}$ and $\{\hat{w}_1, \dots, \hat{w}_m\}$ following equation (3), where $\pi_j \in \Delta^{d_a}$, $\hat{w}_j \geq 0 \; \forall j \in \{1, \dots, m\}$ and $\sum_{j=1}^{m} \hat{w}_j = 1$. If the entropy of $\pi$ is maximized and $\|\pi_1\|_\infty \ll \|\pi_2\|_\infty, \|\pi_1\|_\infty \ll \|\pi_3\|_\infty, \dots, \|\pi_1\|_\infty \ll \|\pi_m\|_\infty$, then $\hat{w}_1 \to 1$.*

*Proof.* Since $\pi \in \Delta^{d_a}$ is defined as equation (3), and its entropy is maximized,

$$\pi = \sum_{j=1}^{m} \hat{w}_j \pi_j = \left[ \frac{1}{d_a}, \dots, \frac{1}{d_a} \right]^\top. \tag{12}$$

Equation (12) holds since $\pi$ becomes a uniform distribution if its entropy is maximized.

Since each $\pi_j$ in equation (12) is a probability simplex, it can be written as

$$\pi_j = \left[ \frac{1}{d_a} + \varepsilon_{j,1}, \frac{1}{d_a} + \varepsilon_{j,2}, \dots, \frac{1}{d_a} + \varepsilon_{j,d_a} \right]^\top, \quad \text{where} \tag{13}$$

$$\sum_{i=1}^{d_a} \varepsilon_{j,i} = 0 \quad \forall j \in \{1, \dots, m\}. \tag{14}$$

Substituting equation (13) into equation (12) and, after some rearrangement, we get

$$\hat{w}_1 \left[ \frac{1}{d_a} + \varepsilon_{1,1}, \dots, \frac{1}{d_a} + \varepsilon_{1,d_a} \right]^\top = \left[ \frac{1}{d_a}, \dots, \frac{1}{d_a} \right]^\top - \sum_{j=2}^{m} \hat{w}_j \left[ \frac{1}{d_a} + \varepsilon_{j,1}, \dots, \frac{1}{d_a} + \varepsilon_{j,d_a} \right]^\top. \tag{15}$$

Without loss of generality, we can assume that

$$\max_{i \in \{1, \dots, d_a\}} \varepsilon_{1,i} = \varepsilon_{1,1} \tag{16}$$

$$\min_{i \in \{1, \dots, d_a\}} \varepsilon_{j,i} = \varepsilon_{j,1} \quad \text{and} \quad \varepsilon_{j,1} \leq 0 \quad \forall j \in \{2, \dots, m\}. \tag{17}$$

Then take the infinity norm for both sides of equation (15), we get

$$\left\| \hat{w}_1 \left[ \frac{1}{d_a} + \varepsilon_{1,1}, \dots, \frac{1}{d_a} + \varepsilon_{1,d_a} \right]^\top \right\|_\infty = \left\| \left[ \frac{1}{d_a}, \dots, \frac{1}{d_a} \right]^\top - \sum_{j=2}^{m} \hat{w}_j \left[ \frac{1}{d_a} + \varepsilon_{j,1}, \dots, \frac{1}{d_a} + \varepsilon_{j,d_a} \right]^\top \right\|_\infty \tag{18}$$

$$\hat{w}_1 \left( \frac{1}{d_a} + \varepsilon_{1,1} \right) = \frac{1}{d_a} - \sum_{j=2}^{m} \hat{w}_j \left( \frac{1}{d_a} + \varepsilon_{j,1} \right). \tag{19}$$

After some rearrangement, equation (19) becomes

$$\frac{1}{d_a} \left( \sum_{j=1}^{m} \hat{w}_j \right) + \hat{w}_1 \varepsilon_{1,1} = \frac{1}{d_a} - \sum_{j=2}^{m} \hat{w}_j \varepsilon_{j,1} \tag{20}$$

$$\hat{w}_1 \varepsilon_{1,1} = - \sum_{j=2}^{m} \hat{w}_j \varepsilon_{j,1}. \tag{21}$$

Equation (21) holds since $\sum_{j=1}^{m} \hat{w}_j = 1$.

Given equation (14), (17), and the assumption that $\|\pi_1\|_\infty \ll \|\pi_2\|_\infty, \|\pi_1\|_\infty \ll \|\pi_3\|_\infty, \dots, \|\pi_1\|_\infty \ll \|\pi_m\|_\infty$, we have the following inequalities

$$\varepsilon_{1,1} \ll -\varepsilon_{j,1} \quad \forall j \in \{1, \dots, m\}. \tag{22}$$

Hence, for equation (21) to hold, $\hat{w}_1 \to 1$. $\qquad\square$

Proposition 4.1 states that if $\|\pi_1\|_\infty$ is much smaller than $\|\pi_j\|_\infty$, then maximizing the entropy of $\pi$ results in $\hat{w}_1 \to 1$ and $\hat{w}_j \to 0, \forall j \in \{2, \dots, m\}$. Next, we provide another proposition showing that if $\pi_1$ is a uniform distribution and $\pi_j$ is not, then $\|\pi_1\|_\infty < \|\pi_j\|_\infty, \forall j \in \{2, \dots, m\}$.

**Proposition A.1** (Infinity norm of a probability simplex). *Given two $d_a$-dimensional probability simplices, $\pi_1 \in \Delta^{d_a}$ and $\pi_2 \in \Delta^{d_a}$, if $\pi_1$ is a uniform distribution and $\pi_2$ is not, then $\|\pi_1\|_\infty = \frac{1}{d_a} < \|\pi_2\|_\infty$.*

*Proof.* Since $\pi_1 \in \Delta^{d_a}$ is a uniform distribution, $\pi_1 = [\frac{1}{d_a}, \frac{1}{d_a}, \ldots, \frac{1}{d_a}]^\top$. The infinity norm of $\pi_1$ becomes

$$\|\pi_1\|_\infty = \max_{i \in \{1,\ldots,d_a\}} |\pi_{1,i}| = \frac{1}{d_a}, \tag{23}$$

where $\pi_{1,i}$ is the $i$-th element of $\pi_1$. On the other hand, since $\pi_2 \in \Delta^{d_a}$ is not a uniform distribution, it can be represented as

$$\pi_2 = \left[\frac{1}{d_a} + \varepsilon_{2,1}, \frac{1}{d_a} + \varepsilon_{2,2}, \ldots, \frac{1}{d_a} + \varepsilon_{2,d_a}\right]^\top, \text{ where} \tag{24}$$

$$\sum_{i=1}^{d_a} \varepsilon_{2,i} = 0 \text{ and } \{\varepsilon_{2,1}, \ldots, \varepsilon_{2,d_a} | \varepsilon_{2,i} \neq 0\} \neq \emptyset. \tag{25}$$

Equation (25) indicates that at least one element in $\{\varepsilon_{2,1}, \ldots, \varepsilon_{2,d_a}\}$ should be larger than 0. Hence,

$$\|\pi_2\|_\infty = \max_{i \in \{1,\ldots,d_a\}} |\pi_{2,i}| = \max_{i \in \{1,\ldots,d_a\}} \left|\frac{1}{d_a} + \varepsilon_{2,i}\right| > \frac{1}{d_a}. \tag{26}$$

$\square$

## A.2   Proof of Proposition 4.2

In continuous action space, the final policy fused by equation (3) is a mixture of normal distributions, also known as a Gaussian mixture. However, in general, the entropy of a Gaussian mixture does not have a closed form [45]. Instead of analyzing the entropy of $\pi$, we analyze the variance of $\pi$ for $d_a = 1$ in maximum entropy KGRL since for any probability density function of a real-valued random variable, Shannon's inequality for entropy and variance [44, 47] specifies

$$H(\pi) \leq \frac{1}{2} \ln\left(2p\sigma^2\right) + \frac{1}{2}, \tag{27}$$

where $\sigma^2 \in \mathbb{R}_{\geq 0}$ is the variance of $\pi$ and $p \approx 3.14159$.

**Proposition 4.2** (Entropy imbalance in continuous control). *Assume a one-dimensional policy distribution $\pi$ is fused by*

$$\pi = \hat{w}_1 \pi_1 + \hat{w}_2 \pi_2, \text{ where } \pi_j = \mathcal{N}(\mu_j, \sigma_j^2), \hat{w}_j \geq 0 \; \forall j \in \{1,2\}, \text{ and } \hat{w}_1 + \hat{w}_2 = 1. \tag{28}$$

*If the variance of $\pi$ is maximized, and $\sigma_1^2 \gg \sigma_2^2$ and $\sigma_1^2 \gg (\mu_1 - \mu_2)^2$, then $\hat{w}_1 \to 1$.*

*Proof.* Let $a$ be a continuous random variable with the probability density function being $\pi(\cdot|\mathbf{s})$. Then its first and second moments are

$$\mathbb{E}_{a \sim \pi}[a] = \int a\pi(\cdot|\mathbf{s}) \, da \tag{29}$$

$$= \int a\left(\hat{w}_1 \pi_1(a|\mathbf{s}) + \hat{w}_2 \pi_2(a|\mathbf{s})\right) da \tag{30}$$

$$= \hat{w}_1 \int a\pi_1(a|\mathbf{s}) \, da + \hat{w}_2 \int a\pi_2(a|\mathbf{s}) \, da \tag{31}$$

$$= \hat{w}_1 \mu_1 + \hat{w}_2 \mu_2 \tag{32}$$

$$\mathbb{E}_{a \sim \pi}\left[a^2\right] = \int a^2 \pi(\cdot|\mathbf{s}) \, da \tag{33}$$

$$= \int a^2 \left(\hat{w}_1 \pi_1(a|\mathbf{s}) + \hat{w}_2 \pi_2(a|\mathbf{s})\right) da \tag{34}$$

$$= \hat{w}_1 \int a^2 \pi_1(a|\mathbf{s}) \, da + \hat{w}_2 \int a^2 \pi_2(a|\mathbf{s}) \, da \tag{35}$$

$$= \hat{w}_1(\mu_1^2 + \sigma_1^2) + \hat{w}_2(\mu_2^2 + \sigma_2^2). \tag{36}$$

Equation (36) holds since $\sigma_j^2 = \mathbb{E}_{a_j \sim \pi_j}[a_j^2] - \left(\mathbb{E}_{a_j \sim \pi_j}[a_j]\right)^2, \forall j \in \{1,2\}$. The variance of $a$ thus becomes

$$\mathbb{V}_{a \sim \pi}[a] = \mathbb{E}_{a \sim \pi}\left[a^2\right] - (\mathbb{E}_{a \sim \pi}[a])^2 \tag{37}$$

$$= \hat{w}_1(\mu_1^2 + \sigma_1^2) + \hat{w}_2(\mu_2^2 + \sigma_2^2) - (\hat{w}_1 \mu_1 + \hat{w}_2 \mu_2)^2 \tag{38}$$

$$= \hat{w}_1 \sigma_1^2 + \hat{w}_2 \sigma_2^2 + \hat{w}_1(1 - \hat{w}_1)\mu_1^2 + \hat{w}_2(1 - \hat{w}_2)\mu_2^2 - 2\hat{w}_1 \hat{w}_2 \mu_1 \mu_2 \tag{39}$$

$$= \hat{w}_1 \sigma_1^2 + \hat{w}_2 \sigma_2^2 + \hat{w}_1 \hat{w}_2(\mu_1^2 + \mu_2^2 - 2\mu_1 \mu_2) \tag{40}$$

$$= \hat{w}_1 \sigma_1^2 + \hat{w}_2 \sigma_2^2 + \hat{w}_1 \hat{w}_2(\mu_1 - \mu_2)^2. \tag{41}$$

According to equation (41), if $\sigma_1^2 \gg \sigma_2^2$ and $\sigma_1^2 \gg (\mu_1 - \mu_2)^2$, maximizing $\mathbb{V}_{a \sim \pi}[a]$ leads to $\hat{w}_1 \to 1$. $\square$

## A.3 Proof of Proposition 4.3

**Proposition 4.3** (Distribution separation in continuous control). *Assume a one-dimensional policy distribution $\pi$ is fused by equation (4). If $\hat{w}_1$, $\hat{w}_2$, $\sigma_1^2$, and $\sigma_2^2$ are fixed, then maximizing the variance of $\pi$ will increase the distance between $\mu_1$ and $\mu_2$.*

*Proof.* According to equation (41), if $\hat{w}_1$, $\hat{w}_2$, $\sigma_1^2$, and $\sigma_2^2$ are fixed, maximizing $\mathbb{V}_{a\sim\pi}[a]$ results in maximizing $(\mu_1 - \mu_2)^2$, hence increasing the distance between $\mu_1$ and $\mu_2$. $\qquad\square$

## A.4 Proof of Proposition 4.4

Before proving Proposition 4.4, we first show that KL divergence between a mixture of uniform distributions and a mixture of normal distributions is upper-bounded by a constant.

**Proposition A.2** (KL divergence between a mixture of uniform distributions and a Gaussian mixture). *Given a Gaussian mixture*

$$\pi(\cdot) = \sum_{j=1}^{m} \hat{w}_j \, \mathcal{N}(\cdot; \mu_j, \sigma_j^2), \tag{42}$$

$$where \quad \mathcal{N}(x; \mu_j, \sigma_j^2) = \frac{1}{\sqrt{2p\sigma_j^2}} \exp \frac{-(x-\mu_j)^2}{2\sigma_j^2} \tag{43}$$

*and a mixture of $m$ uniform distributions*

$$\hat{\pi}(\cdot) = \sum_{j=1}^{m} \hat{w}_j \, \mathcal{U}(\cdot; \mu_j - \gamma_j, \mu_j + \gamma_j), \tag{44}$$

$$where \quad \mathcal{U}(\cdot; a, b) = \frac{1}{b-a} \quad and \quad \gamma_j = \frac{1}{2\,\mathcal{N}(\mu_j; \mu_j, \sigma_j^2)} \tag{45}$$

*for a real-valued random variable, the KL divergence between $\hat{\pi}(\cdot)$ and $\pi(\cdot)$ has an upper bound of $\frac{p}{12}$.*

*Proof.* Since KL divergence $D_{KL}(P\|Q)$ is convex in the pair $(P, Q)$, $D_{KL}(\hat{\pi}\|\pi)$ has the following upper bound [43]

$$D_{KL}(\hat{\pi}\|\pi) = D_{KL}\left( \sum_{j=1}^{m} \hat{w}_j \, \mathcal{U}(\cdot; \mu_j - \gamma_j, \mu_j + \gamma_j) \,\Big\|\, \sum_{j=1}^{m} \hat{w}_j \, \mathcal{N}(\cdot; \mu_j, \sigma_j^2) \right) \tag{46}$$

$$\leq \sum_{j=1}^{m} \hat{w}_j D_{KL}\left( \mathcal{U}(\cdot; \mu_j - \gamma_j, \mu_j + \gamma_j) \,\|\, \mathcal{N}(\cdot; \mu_j, \sigma_j^2) \right). \tag{47}$$

For each $j \in \{1, \ldots, m\}$,

$$D_{KL}\left( \mathcal{U}(\cdot; \mu_j - \gamma_j, \mu_j + \gamma_j) \,\|\, \mathcal{N}(\cdot; \mu_j, \sigma_j^2) \right) \tag{48}$$

$$= \int_{\mu_j-\gamma_j}^{\mu_j+\gamma_j} \frac{1}{2\gamma_j} \ln \frac{\frac{1}{2\gamma_j}}{\frac{1}{\sqrt{2p\sigma_j^2}} \exp\left( -\frac{(x-\mu_j)^2}{2\sigma_j^2} \right)} \, dx \tag{49}$$

$$= \int_{\mu_j-\gamma_j}^{\mu_j+\gamma_j} \frac{1}{2\gamma_j} \ln \frac{1}{2\gamma_j} \, dx - \int_{\mu_j-\gamma_j}^{\mu_j+\gamma_j} \frac{1}{2\gamma_j} \left( \ln \frac{1}{\sqrt{2p\sigma_j^2}} - \frac{(x-\mu_j)^2}{2\sigma_j^2} \right) dx \tag{50}$$

$$= \ln \frac{1}{2\gamma_j} - \ln \frac{1}{\sqrt{2p\sigma_j^2}} + \int_{\mu_j-\gamma_j}^{\mu_j+\gamma_j} \frac{(x-\mu_j)^2}{4\gamma_j\sigma_j^2} \, dx \tag{51}$$

$$= \ln \frac{1}{2\gamma_j} - \ln \frac{1}{\sqrt{2p\sigma_j^2}} + \frac{\gamma_j^2}{6\sigma_j^2} \tag{52}$$

$$= \frac{p}{12}. \tag{53}$$

Equation (53) comes from substituting $\gamma_j$ into equation (52).

Finally, the upper bound of $D_{KL}(\hat{\pi}\|\pi)$ becomes

$$D_{KL}(\hat{\pi}\|\pi) \le \sum_{j=1}^{m} \hat{w}_j D_{KL}\left(\mathcal{U}(\cdot; \mu_j - \gamma_j, \mu_j + \gamma_j) \,\|\, \mathcal{N}(\cdot; \mu_j, \sigma_j^2)\right) \tag{54}$$

$$= \sum_{j=1}^{m} \hat{w}_j \frac{p}{12} = \frac{p}{12}. \tag{55}$$

$\square$

**Proposition 4.4** (Approximation of a mixture of normal distributions). *If the following three inequalities hold for $\mu_{t,in}, \mu_{t,g_1}, \dots, \mu_{t,g_n}$, and $a_{t,in}$: $\|\mu_{t,in} - \mu_{t,g_j}\|_2 < \min\{\gamma_{t,in}, \gamma_{t,g_j}\}$, $\|a_{t,in} - \mu_{t,in}\|_2 < \min\{\gamma_{t,in}, \gamma_{t,g_j}\}$, and $\|a_{t,in} - \mu_{t,g_j}\|_2 < \gamma_{t,g_j}$, $\forall j \in \{1, \dots, n\}$, where $\gamma_{t,in} = 1/(2\pi_{in}(\mu_{t,in}|\mathbf{s}_t))$ and $\gamma_{t,g_j} = 1/(2\pi_{g_j}(\mu_{t,g_j}|\mathbf{s}_t))$, then equation (9) for a real-valued action $a_t$ sampled from KIAN can be approximated by*

$$\hat{w}_{t,in}\mathcal{U}(a_t; \mu_{t,in} - \gamma_{t,in}, \mu_{t,in} + \gamma_{t,in}) + \sum_{j=1}^{n} \hat{w}_{t,g_j}\mathcal{U}(a_t; \mu_{t,in} - \gamma_{t,g_j}, \mu_{t,in} + \gamma_{t,g_j}), \tag{56}$$

*where $\quad \mathcal{U}(\cdot; a, b) = 1/(b - a).$* $\tag{57}$

*In addition, equation (8) is a lower bound of equation (10).*

*Proof.* Proposition A.2 shows that $\pi(\cdot|\mathbf{s}_t)$ fused as equation (3) can be approximated by

$$\hat{\pi}(\cdot|\mathbf{s}_t) = \hat{w}_{t,in}\mathcal{U}(\cdot; \mu_{t,in} - \gamma_{t,in}, \mu_{t,in} + \gamma_{t,in}) + \sum_{j=1}^{n} \hat{w}_{t,g_j}\mathcal{U}(\cdot; \mu_{t,g_j} - \gamma_{t,g_j}, \mu_{t,g_j} + \gamma_{t,g_j}) \tag{58}$$

with KL divergence being at most $\frac{p}{12}$, which is a constant.

Since any continuous action $a_t$ outputted by KIAN belongs to $\{a_{t,in}, \mu_{t,g_1}, \dots, \mu_{t,g_n}\}$ (Line 11 to 16 in Algorithm 1), if the three inequalities in the proposition statement hold, then for all $a_t$

$$\mathcal{U}(a_t; \ \mu_{t,in} - \gamma_{t,in}, \mu_{t,in} + \gamma_{t,in}) = \frac{1}{2\gamma_{t,in}}, \tag{59}$$

$$\mathcal{U}(a_t; \ \mu_{t,g_j} - \gamma_{t,g_j}, \mu_{t,g_j} + \gamma_{t,g_j}) = \frac{1}{2\gamma_{t,g_j}} \quad \forall j \in \{1, \dots, n\}, \text{ and} \tag{60}$$

$$\mathcal{U}(a_t; \ \mu_{t,in} - \gamma_{t,g_j}, \mu_{t,in} + \gamma_{t,g_j}) = \frac{1}{2\gamma_{t,g_j}} \quad \forall j \in \{1, \dots, n\}. \tag{61}$$

Therefore, $\hat{\pi}(\cdot|\mathbf{s}_t)$ can be rewritten as

$$\hat{\pi}(\cdot|\mathbf{s}_t) = \hat{w}_{t,in}\mathcal{U}(\cdot; \mu_{t,in} - \gamma_{t,in}, \mu_{t,in} + \gamma_{t,in}) + \sum_{j=1}^{n} \hat{w}_{t,g_j}\mathcal{U}(\cdot; \mu_{t,in} - \gamma_{t,g_j}, \mu_{t,in} + \gamma_{t,g_j}). \tag{62}$$

For any $a_t \in \{a_{t,in}, \mu_{t,g_1}, \dots, \mu_{t,g_n}\}$,

$$\hat{\pi}(a_t|\mathbf{s}_t) = \hat{w}_{t,in}\mathcal{U}(a_t; \mu_{t,in} - \gamma_{t,in}, \mu_{t,in} + \gamma_{t,in}) + \sum_{j=1}^{n} \hat{w}_{t,g_j}\mathcal{U}(a_t; \mu_{t,in} - \gamma_{t,g_j}, \mu_{t,in} + \gamma_{t,g_j}) \tag{63}$$

$$= \hat{w}_{t,in}\frac{1}{2\gamma_{t,in}} + \sum_{j=1}^{n} \hat{w}_{t,g_j}\frac{1}{2\gamma_{t,g_j}} \tag{64}$$

$$= \hat{w}_{t,in}\pi_{in}(\mu_{t,in}|\mathbf{s}_t) + \sum_{j=1}^{n} \hat{w}_{t,g_j}\pi_{g_j}(\mu_{t,g_j}|\mathbf{s}_t) \tag{65}$$

$$\ge \hat{w}_{t,in}\pi_{in}(a_{t,in}|\mathbf{s}_t) + \sum_{j=1}^{n} \hat{w}_{t,g_j}\pi_{g_j}(\mu_{t,g_j}|\mathbf{s}_t). \tag{66}$$

Inequality (66) holds since $\pi_{in}(a_t|\mathbf{s}_t)$ has a maximum value when $a_t = \mu_{t,in}$, and it shows that equation (8) is a lower bound of equation (10). $\square$

We use equation (8) instead of (10) to approximated equation (9) since it includes more information about the distribution $\pi_{in}(\cdot|\mathbf{s}_t)$ and helps adjust the learnable variance $\sigma_{t,in}^2$.

## A.5 Proof of Proposition 4.5

**Proposition 4.5** (Maximized variance's independence of the distance between means)**.** *Assume a one-dimensional policy $\pi$ is fused by equation (4). If $\pi(a|\mathbf{s})$ is approximated as equation (8), and the three inequalities in Proposition 4.4 are satisfied, then maximizing the variance of $\pi(\cdot|\mathbf{s})$ will not affect the distance between $\mu_1$ and $\mu_2$.*

*Proof.* Proposition 4.4 shows that approximating $\pi(a|\mathbf{s})$ as equation (8) comes from approximating $\pi(\cdot|\mathbf{s})$ with

$$\hat{\pi}(\cdot|\mathbf{s}) = \hat{w}_1 \mathcal{U}(\cdot; \mu_1 - \gamma_1, \mu_1 + \gamma_1) + \hat{w}_2 \mathcal{U}(\cdot; \mu_1 - \gamma_2, \mu_1 + \gamma_2). \tag{67}$$

Let $a$ be a continuous random variable with the probability density function being $\hat{\pi}(\cdot|\mathbf{s})$. Following the proof of Proposition 4.2, its first and second moments are

$$\mathbb{E}_{a \sim \hat{\pi}}[a] = \mu_1 \tag{68}$$

$$\mathbb{E}_{a \sim \hat{\pi}}[a^2] = \mu_1^2 + \hat{w}_1 \frac{\gamma_1^2}{3} + \hat{w}_2 \frac{\gamma_2^2}{3}. \tag{69}$$

Equation (69) holds since the variance of $\mathcal{U}(\cdot; a, b)$ is $\frac{(b-a)^2}{12}$. Then the variance of $a$ becomes

$$\mathbb{V}_{a \sim \hat{\pi}}[a] = \hat{w}_1 \frac{\gamma_1^2}{3} + \hat{w}_2 \frac{\gamma_2^2}{3} \tag{70}$$

$$= \frac{p}{6} (\hat{w}_1 \sigma_1^2 + \hat{w}_2 \sigma_2^2), \tag{71}$$

which is not related to the distance between $\mu_1$ and $\mu_2$. $\qquad\square$

## A.6 Learning Algorithms for KIAN

---

**Algorithm 1:** Knowledge-Grounded RL with KIAN

---

**Input:** environment $\mathcal{E}$ with a KGMDP $(\mathcal{S}, \mathcal{A}, \mathcal{T}, \mathcal{G}, \mathcal{R}, \rho, \gamma)$, where $\mathcal{G} = \{\pi_{g_1}, \pi_{g_2}, \ldots, \pi_{g_n}\}$

1 Initialize $\boldsymbol{\theta}, \boldsymbol{\phi}$, and $\mathbf{k}_e, \forall e \in \{in, g_1, \ldots, g_n\}$
2 Observe a state $\mathbf{s}_0 \in \mathcal{S}$ from $\mathcal{E}$
3 **for** *each time step $t$* **do**
     // Compute weights for all knowledge policies
4   | **if** $\mathcal{A}$ *is a discrete action space* **then**
5   |  | Compute $w_{in}, w_{g_1}, \ldots, w_{g_n}$ according to equation (6)
6   | **else if** $\mathcal{A}$ *is a continuous action space* **then**
7   |  | Compute $w_{in}, w_{g_1}, \ldots, w_{g_n}$ according to equation (1) with
    |  | $c_{t,e} = 1, \forall e \in \{in, g_1, \ldots, g_n\}$
8   | Compute $\hat{w}_{in}, \hat{w}_{g_1}, \ldots, \hat{w}_{g_n}$ according to (2)
    | // Sample an action
9   | **if** $\mathcal{A}$ *is a discrete action space* **then**
10   |  | Sample an action $\mathbf{a}_t \sim \pi(\cdot|\mathbf{s}_t)$, with $\pi$ following equation (5)
11   | **else if** $\mathcal{A}$ *is a continuous action space* **then**
12   |  | Sample a knowledge policy $e \sim \texttt{gumbel\_softmax}([\hat{w}_{t,in}, \hat{w}_{t,g_1}, \ldots, \hat{w}_{t,g_n}]^\top)$
13   |  | **if** $e = in$ **then**
14   |  |  | Sample an action $\mathbf{a}_t \sim \pi_{in}(\cdot|\mathbf{s}_t)$
15   |  | **else**
16   |  |  | $\mathbf{a}_t = \boldsymbol{\mu}_{e,t}$
    | // Apply the action to the environment
17   | Apply $\mathbf{a}_t \in \mathcal{A}$ to $\mathcal{E}$ and observe a reward $R_t$ and the next state $\mathbf{s}_{t+1} \in \mathcal{S}$
    | // Update KIAN
18   | **if** $\mathcal{A}$ *is a discrete action space* **then**
19   |  | Compute entropy-related term with $\pi(\cdot|\mathbf{s}_t)$ following equation (5)
20   | **else if** $\mathcal{A}$ *is a continuous action space* **then**
21   |  | Compute entropy-related term with $\pi(\mathbf{a}_t|\mathbf{s}_t)$ following equation (8)
22   | Update $\boldsymbol{\theta}, \boldsymbol{\phi}$, and $\mathbf{k}_{in}, \mathbf{k}_{g_1}, \ldots, \mathbf{k}_{g_n}$ with any (maximum entropy) RL algorithm

**Output:** $\boldsymbol{\theta}, \boldsymbol{\phi}$, and $\mathbf{k}_{in}, \mathbf{k}_{g_1}, \ldots, \mathbf{k}_{g_n}$

---

# B  Experimental Details

All experiments are conducted using Pytorch [22].

## B.1  Baseline Algorithms

We compare KIAN with the following baselines that incorporate external knowledge policies differently.

- **Behavior cloning (BC)** [3]: An agent follows only the policies in $\mathcal{G}$ to solve a task. This method is the optimal solution for supervised learning from demonstrations, where external knowledge policies generate the demonstrations.
- **RL** [10, 29]: An agent learns a policy by RL without any external guidance.
- **RL+BC** [21]: An agent learns a policy by RL with BC signals integrated. These signals come from demonstrations generated by external knowledge policies.
- **KoGuN** [36]: An agent learns a policy with the input being a concatenation of a state and $n$ actions suggested by all policies in $\mathcal{G}$.
- **A2T** [27]: An agent fuses a learnable inner policy with $n$ external policies in $\mathcal{G}$ by learning a function approximator that predicts $n + 1$ weights for all policies.

## B.2  MiniGrid Environments

### B.2.1  Environmental Details

We evaluate all methods on the following tasks in MiniGrid environments (`https://github.com/maximecb/gym-minigrid`): Empty-Random-5x5, Unlock, DoorKey-5x5, DoorKey-8x8, Dynamic-Obstacles-16x16, LavaCrossingS9N2, MultiRoom-N4-S5, and KeyCorridorS3R2. A state $\mathbf{s}_t$ in each task is a directed first-person view represented as a 5x5 grid. An action $\mathbf{a}_t$ in each task is one of the six discrete actions: left, right, forward, pickup, drop, and toggle.

### B.2.2  Initial External Knowledge Set

The initial external knowledge set, $\mathcal{G}^{init}$, for MiniGrid tasks comprises eight sub-optimal if-else-based programs, such as:

- `pick_up_the_key`: If there exists a key in $s_t$, move to the key; if the key is in front of the agent, $\pi_{g_j}(pickup|s_t) = 1$.
- `pick_up_the_ball`: If there exists a ball in $s_t$, move to the key; if the key is in front of the agent, $\pi_{g_j}(pickup|s_t) = 1$.
- `open_the_door`: If there exists a door in $s_t$, move to the door; if the door is in front of the agent, $\pi_{g_j}(toggle|s_t) = 1$.
- `open_the_locked_door`: If there exists a locked door in $s_t$, move to the door; if the door is in front of the agent, $\pi_{g_j}(toggle|s_t) = 1$.
- `open_the_unlocked_door`: If there exists a unlocked door in $s_t$, move to the door; if the door is in front of the agent, $\pi_{g_j}(toggle|s_t) = 1$.
- `go_to_the_goal`: If there exists a goal in $s_t$, move to the goal.
- `do_not_hit`: If there exists walls or lava around the agent, do not choose the direction.
- `do_not_hit_balls`: If there exists balls around the agent, do not choose the direction.

In the above policies, the 'move to' is decided by $p_{obj} - p_{self}$, where $\mathbf{p}_{self} \in \mathbb{R}^3$ is the position of the agent and $\mathbf{p}_{obj} \in \mathbb{R}^3$ is the position of the object. The $\pi_{g_j}$ of the actions right, left, and forward, can be written as:

$$\begin{cases} \pi_{g_j}(right|s_t) = 1, & \text{if } \mathbf{p}_{obj,x} - \mathbf{p}_{self,x} > 0 \\ \pi_{g_j}(left|s_t) = 1, & \text{if } \mathbf{p}_{obj,x} - \mathbf{p}_{self,x} < 0 \\ \pi_{g_j}(forward|s_t) = 1, & \text{if } \mathbf{p}_{obj,y} - \mathbf{p}_{self,y} > 0 \end{cases} \tag{72}$$

### B.2.3  Model Architecture

The eight environments in Minigrid share the same model architecture. Each method involves learning an image encoding, an actor and a critic networks. The architecture of the image encoding network and critic network are the same for all methods, but their actor networks have different architectures.

**Image Encoding Network.** The image encoding network is a three-layer convolutional neural network that maps an input image to an image embedding, which is used as the state by actor and critic networks.

**Critic Network.** A critic network is a multi-layer perceptron (MLP) that predicts a state value [29]. The architecture of a critic network has one hidden layer that contains 64 units. Each hidden layer is followed by Tanh activation.

**Actor Network of PPO, PPO+BC, and KoGuN.** An actor network of PPO, PPO+BC and KoGuN is an MLP with one hidden layers and a hidden size of 64 units.

**Actor Network of A2T.** An actor network of A2T contains an internal actor network and an attention network. The internal actor network has the same architecture as an actor of PPO, PPO+BC, and KoGuN. The attention network is an MLP with one hidden layer and a hidden size of 64 units base.

**KIAN.** The internal actor network of KIAN has the same architecture as an actor of PPO, PPO+BC, and KoGuN. Each knowledge key is a learnable vector with $d_k = 8$ and modeled by the PyTorch module, `nn.Embedding`. The query network is an additional output layer projecting 64-dim to $d_k$-dim that based on the inner actor's first layer outputs.

### B.2.4 Hyperparameters

We implement all methods based on the implementation of PPO in `https://github.com/lcswillems/rl-starter-files`. The training timesteps are 75K, 300K, 75K, 3M, 300K, 5M, 500K, 1M for Empty-Random-5x5, Unlock, DoorKey-5x5, DoorKey-8x8, Dynamic-Obstacles-16x16, LavaCrossingS9N2, MultiRoom-N4-S5, and KeyCorridorS3R2 respectively. The learning rates are $1 \times 10^{-3}$ for all tasks. The batch sizes are 256 for all tasks. The discount factors $\gamma = 0.99$ for all tasks. The coefficient of the entropy term $\alpha$ is searched to be 0 or 0.01, the default value.

## B.3 OpenAI-Robotic Environments

### B.3.1 Environmental Details

We evaluate all methods on the following tasks in OpenAI-Robotic environments: FetchPush, FetchSlide, and FetchPickAndPlace. A state $\mathbf{s}_t \in \mathbb{R}^{25}$ in each task contains (1) the position and velocity of the end-effector, (2) the position, rotation, and velocity of the object, (3) the relative position between the object and the end-effector, and (4) the distance between the two grippers and their velocity. An action $\mathbf{a}_t \in \mathbb{R}^4$ in each task contains the position variation of the end-effector and the distance between the two grippers.

### B.3.2 Initial External Knowledge Set

The initial external knowledge set, $\mathcal{G}^{init}$, for all OpenAI-Robotic tasks comprises two sub-optimal if-else-based programs, `move_forward_to_the_object` and `move_forward_to_the_goal`.

- `move_forward_to_the_object`: If $\|\mathbf{p}_{ee} - \mathbf{p}_{obj}\|_2 \geq \varepsilon$, move straightly to the object with the gripper opened; otherwise, stay unmoved.
- `move_forward_to_the_goal`: If $\|\mathbf{p}_{ee} - \mathbf{p}_{obj}\|_2 < \varepsilon$, move straightly to the goal with the gripper closed; otherwise, stay unmoved.

In the above two policies, $\mathbf{p}_{ee} \in \mathbb{R}^3$ is the position of the end-effector, and $\mathbf{p}_{obj} \in \mathbb{R}^3$ is the position of the object. For all tasks, $\varepsilon = 0.03$.

### B.3.3 Model Architecture

FetchPush, FetchSlide, and FetchPickAndPlace share the same model architecture. Each method involves learning an actor and a critic network. The architecture of the critic network is the same for all methods, but their actor networks have different architectures.

**Critic Network.** A critic network is a multi-layer perceptron (MLP) that predicts a state-action value [**?** ]. The architecture of a critic network has three hidden layers, and each layer contains 512 units. Each hidden layer is followed by ReLU activation.

**Actor Network of SAC, SAC+BC, and KoGuN.** An actor network of SAC, SAC+BC and KoGuN is an MLP with three hidden layers and a hidden size of 512 units.

**Actor Network of A2T.** An actor network of A2T contains an internal actor network and an attention network. The internal actor network has the same architecture as an actor of SAC, SAC+BC, and KoGuN. The attention network is an MLP with two hidden layers and a hidden size of 64 units.

**KIAN.** The internal actor network of KIAN has the same architecture as an actor of SAC, SAC+BC, and KoGuN. Each knowledge key is a learnable vector with $d_k = 4$ and modeled by the PyTorch module, `nn.Embedding`. The query network is an MLP with two hidden layers and a hidden size of 64 units.

### B.3.4 Hyperparameters

We implement all methods based on the implementation of SAC in Stable-Baselines3 (SB3) [26]. The training timesteps are 1M for FetchPush and FetchPickAndPlace and 1.2M for FetchSlide. The learning rates are $5 \times 10^{-4}$ for FetchPush and FetchPickAndPlace and $4 \times 10^{-4}$ for FetchSlide. The batch sizes are 2048 for all tasks. The replay-buffer sizes are 1M for all tasks. The discount factors $\gamma = 0.95$ for all tasks. The coefficient of the entropy term $\alpha$ is adjusted automatically for all tasks as described in [10].

## C  Details of Compositional and Incremental Experiments

### C.1  MiniGrid Environments

After learning an actor in $\mathcal{M}_k^1$ with the experimental setup described in Section B, we train KIAN for $\mathcal{M}_k^2$ by initializing its external knowledge keys with the knowledge keys learned in $\mathcal{M}_k^1$. These external knowledge keys remain fixed when learning $\mathcal{M}_k^2$, and all other components of KIAN are learned from scratch. This setup allows us to test the efficacy of reusing learned knowledge keys across different tasks. All actor components of RL, KoGuN, and A2T are learned from scratch in $\mathcal{M}_k^2$. This is because RL does not incorporate any external knowledge, and the model architectures of KoGuN and A2T do not allow changing the number and order of knowledge policies. The hyperparameters of learning $\mathcal{M}_k^1$ and $\mathcal{M}_k^2$ are the same as those listed in Section B.2.4. The external knowledge policies used for each experiment are detailed as follows:

- **Dynamic-Obstacles-16x16 → LavaCrossingS9N2.** Reuse the knowledge embedding of "go to the goal"; reuse the knowledge embedding of "do not hit balls" as the fixed knowledge embedding of "do not hit".

- **Unlock → DoorKey-8x8.** Reuse the knowledge embeddings of "get the key" and "open the door".

- **Dynamic-Obstacles-16x16 → MultiRoom-N4-S5.** Reuse the knowledge embedding of "go to the goal".

- **DoorKey-8x8 → MultiRoom-N4-S5.** Reuse the knowledge embedding of "go to the goal"; reuse the knowledge embedding of "open the door" as the fixed knowledge embedding of "open the unlocked door".

- **Dynamic-Obstacles-16x16 → KeyCorridorS3R2.** Reuse the knowledge embedding of "go to the goal" as the fixed knowledge embedding of "pick up the ball".

- **DoorKey-8x8 → KeyCorridorS3R2.** Reuse the knowledge embedding of "get the key"; reuse the knowledge embedding of "open the door"; reuse the knowledge embedding of "go to the goal" as the fixed knowledge embedding of "pick up the ball".

### C.2  OpenAI-Robotic Environments

After learning an actor and a critic in $\mathcal{M}_k^1$ with the experimental setup described in Section B, we initialize the networks for $\mathcal{M}_k^2$ as follows:

- **RL**: The actor and critic of $\mathcal{M}_k^2$ are initialized with that of $\mathcal{M}_k^1$. These networks will be updated when learning in $\mathcal{M}_k^2$.

- **KoGuN**: Only the critic of $\mathcal{M}_k^2$ is initialized with that of $\mathcal{M}_k^1$. The actor is learned from scratch in $\mathcal{M}_k^2$. The actor and critic will be updated when learning in $\mathcal{M}_k^2$.

- **A2T**: The critic of $\mathcal{M}_k^2$ is initialized with that of $\mathcal{M}_k^1$. The inner policy and attention network are learned from scratch in $\mathcal{M}_k^2$. All networks will be updated when learning in $\mathcal{M}_k^2$.

- **KIAN**: The external knowledge keys, query, and critic of $\mathcal{M}_k^2$ are initialized with that of $\mathcal{M}_k^1$. Note that the external knowledge keys of $\mathcal{M}_k^2$ include *the internal and external knowledge keys from $\mathcal{M}_k^1$*. The inner policy and inner knowledge key are learned from scratch in $\mathcal{M}_k^2$. The external knowledge keys remain fixed, while other networks will be updated when learning in $\mathcal{M}_k^2$.

The hyperparameters of learning $\mathcal{M}_k^1$ are the same as those listed in Section B.3.4. When learning $\mathcal{M}_k^2$, the hyperparameters changed are listed as follows: The training timesteps are 0.4M. The learning rates are $7.5 \times 10^{-4}$ and $10^{-3}$ for FetchPush and FetchPickAndPlace respectively.

## D   Broader Impact and Future Research Directions

The KGRL framework presented in this work aims to enhance an agent's ability to learn from external policies. These policies encompass not only sub-optimal strategies to task completion but also regulative policies that emphasize safety constraints and ethical behaviors. Being able to incorporate safety- and ethics-oriented policies gives KGRL the potential to significantly influence an artificial agent's behavior, promoting enhanced safety and social acceptability. These aspects have gained substantial attention in the field of RL [38–42, 46, 48], underscoring their importance in contemporary research.

Moving forward, there are several research directions in KGRL that are worth exploring. First, fusing knowledge policies with different state and action spaces enables efficient learning across a broader range of applications. Second, integrating regulative policies that enforce strict constraints during learning and inference stages can ensure adherence to safety and ethical considerations. Lastly, addressing complex relationships among different policies, such as conditional dependence and conflicts, allows an agent to efficiently navigate through a large and diverse set of knowledge policies. We hope these directions have the potential to inspire future studies in KGRL.

## E   Other Experimental Results

We provide the standard deviation for the experiments of generalizability in Table 3-5.

| Train in
Test in | Empty-Random-5x5 | | | DoorKey-5x5 | |
|---|---|---|---|---|---|
| | 6x6 | 8x8 | 16x16 | 8x8 | 16x16 |
| RL [10, 29] | .88±.06 | .71±.20 | .45±.35 | .29±.16 | .08±.10 |
| RL+BC [21] | .87±.03 | .60±.14 | .24±.17 | .40±.01 | .09±.08 |
| KoGuN [36] | .94±.01 | .83±.03 | .53±.11 | .77±.09 | .35±.08 |
| A2T [27] | .92±.01 | .78±.11 | .51±.30 | .53±.09 | .11±.03 |
| KIAN (ours) | .96±.02 | .91±.01 | .93±.02 | .76±.01 | .42±.08 |

Table 3: MiniGrid Zero-Shot S2C Experiments.

| Train in
Test in | Push | | Slide | | Pick-and-Place | |
|---|---|---|---|---|---|---|
| | 5x | 10x | 5x | 10x | 5x | 10x |
| RL [10, 29] | .87±.05 | .52±.11 | .45±.07 | .17±.05 | .34±.54 | .27±.44 |
| RL+BC [21] | .89±.02 | .60±.09 | .44±.10 | .16±.02 | .34±.55 | .30±.50 |
| KoGuN [36] | .63±.49 | .43±.32 | .55±.07 | .18±.04 | .32±.52 | .24±.38 |
| A2T [27] | .03±.00 | .05±.00 | .00±.00 | .01±.00 | .01±.00 | .06±.00 |
| KIAN (ours) | .93±.05 | .70±.02 | .42±.10 | .15±.04 | .92±.00 | .72±.03 |

Table 4: OpenAI-Robotics Zero-Shot S2C Experiments.

| Train in
Test in | DoorKey-5x5
Empty-Random | DoorKey-8x8
Unlock | 
DoorKey5x5 | Pick-and-Place
Reach | 
Push | Push
Reach | Slide
Push |
|---|---|---|---|---|---|---|---|
| RL [10, 29] | .83±.07 | .92±.01 | .93±.01 | .80±.45 | .31±.19 | .16±.10 | .09±.04 |
| RL+BC [21] | .85±.05 | .87±.03 | .93±.01 | .80±.45 | .31±.19 | .16±.10 | .09±.04 |
| KoGuN [36] | .90±.02 | .91±.01 | .93±.01 | .45±.37 | .05±.02 | .20±.07 | .07±.02 |
| A2T [27] | .84±.04 | .92±.01 | .93±.00 | .01±.00 | .05±.00 | .20±.45 | .05±.00 |
| KIAN (ours) | .91±.01 | .94±.01 | .95±.00 | 1.0±.00 | .30±.04 | .24±.06 | .13±.02 |

Table 5: Zero-Shot C2S Experiments.

## F   Effects of Extensive External Knowledge Set with Irrelevant Policies

In order for KGRL agents to effectively leverage an external knowledge set, it is imperative that they can (1) distinguish which external policies are less related to the given task and (2) efficiently navigate through an

extensive collection of external policies. Failure to accomplish these objectives in a timely manner could result in suboptimal performance, potentially even inferior to that of an RL agent. Under such circumstances, integrating external policies into the learning process becomes impractical.

In this section, we examine the following two aspects of KIAN: (1) the ability of KIAN to rapidly acquire valuable strategies, even in the presence of random or irrelevant policies within the external knowledge set, and (2) the impact of KIAN's performance as the size of the external knowledge set increases.

Figure 6 shows the learning curves of PPO and PPO+KIAN, considering various numbers of external policies: 2 relevant, 4 (2 relevant + 2 irrelevant), and 6 (2 relevant + 4 irrelevant) for the MiniGrid Unlock task. The results indicate that including more irrelevant knowledge policies leads to a marginal decline in performance, but the agents consistently achieve high rewards with minimal variances. This minor decline in performance aligns with our expectations since the agents need to (1) navigate through a more extensive set of external policies and (2) distinguish and disregard policies that do not contribute to solving the task. Therefore, when the external knowledge set is very large, KGRL methods, such as KoGuN, A2T, and KIAN, do not guarantee superior efficiency over RL methods. Efficiently harnessing a large external policy set remains an avenue of future research and exploration.

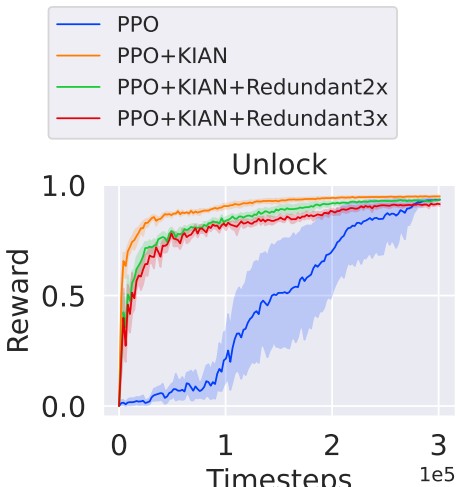

Figure 6: Learning curves of PPO and PPO+KIAN for the MiniGrid Unlock task, with external knowledge sets that include irrelevant policies.

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
