# OpenReview forum: "Flexible Attention-Based Multi-Policy Fusion for Efficient Deep Reinforcement Learning"
_NeurIPS.cc/2023/Conference — NeurIPS 2023 poster_

### Official Review · Reviewer_K32E · 2023-07-02

**Soundness:** 4 excellent
**Presentation:** 3 good
**Contribution:** 3 good
**Rating:** 6
**Confidence:** 4

**Summary:**

This paper introduces KIAN, a method for leveraging multiple sub-optimal policies to solve reinforcement learning tasks. The paper starts out by introducing a new Knowledge-Grounded MDP, which adds a set of external knowledge policies to the traditional MDP framework.

To leverage these external knowledge policies, KIAN learns an embedding for each knowledge policy (including an inner knowledge policy for the current agent). Then at each step, the current state is mapped to a query embedding that can be used to find the policy that is best equipped to take an action at that timestep.

**Strengths:**

* The paper is structured well and is easy to read.
* The authors demonstrated great attention to detail by motivating the theorems and equations with intuition before defining them concretely.
*  The paper is very technically sound.

**Weaknesses:**

* > In such a case, whenever the knowledge set is updated by adding or replacing policies, prior methods require relearning the entire multi-policy fusion process, even if the current task is similar to the previous one. This is because their designs of knowledge representations are intertwined with the knowledge-fusing mechanism, which restricts the number of policies in the knowledge set from being changed.

    * It would be good to point to specific prior works that suffer from this problem so the reader can build intuition.

* There are no visualizations of the tasks used for evaluation. Adding pictures of the environments would help readers understand what the agent needs to do.

**Questions:**

*  I don't fully understand the incremental property. In the introduction. the authors state that "Humans do not need to relearn how to navigate the entire knowledge set from scratch when they remove outdated strategies or add new ones." But then Definition 3.5 states that an incremental agent has two properties:
    1. The agent can use different knowledge policies to solve a single KGMDP
    2. Given a sequence of KGMDPs, the agent can solve them with different knowledge sets.

    * In its current form, definition 3.5 does not specify the case where an agent adds or removes a specific policy from its knowledge set. I think a condition should be added to highlight how an incremental agent should be able to leverage a sequence of knowledge sets that differ by one or more policies.


* How do the initial external knowledge policies perform on the given tasks?

**Limitations:**

The authors discussed broader impact and future work in Appendix D, but no limitations were addressed. Could the authors please add limitations to either Appendix D or the conclusion?

---

> ### Author Rebuttal · Authors · 2023-08-10
>
> 1. *"In such a case, whenever the knowledge set is updated by adding or replacing policies, prior methods require relearning the entire multi-policy fusion process, even if the current task is similar to the previous one. This is because their designs of knowledge representations are intertwined with the knowledge-fusing mechanism, which restricts the number of policies in the knowledge set from being changed."*
> *It would be good to point to specific prior works that suffer from this problem so the reader can build intuition.*
>
> Thank you for the suggestion! We will add citations (e.g., for KoGuN and A2T) after the statement.
>
> 2. *There are no visualizations of the tasks used for evaluation. Adding pictures of the environments would help readers understand what the agent needs to do.*
>
> Thank you for the suggestion! We provided some screenshots of the environments used for evaluation in Figures 6 and 10 in the additional-result document and will add them to the final paper.
>
> 3. *I don't fully understand the incremental property. In the introduction. the authors state that "Humans do not need to relearn how to navigate the entire knowledge set from scratch when they remove outdated strategies or add new ones." But then Definition 3.5 states that an incremental agent has two properties:*
>
> 3.a) The agent can use different knowledge policies to solve a single KGMDP
>
> 3.b) Given a sequence of KGMDPs, the agent can solve them with different knowledge sets.
>
> *In its current form, definition 3.5 does not specify the case where an agent adds or removes a specific policy from its knowledge set. I think a condition should be added to highlight how an incremental agent should be able to leverage a sequence of knowledge sets that differ by one or more policies.*
>
> Thank you for the suggestion! We assumed changing $\mathcal{G}$ includes the operations of adding/removing knowledge policies. We will add concrete conditions of these operations in Definition 3.5.
>
> 4. *How do the initial external knowledge policies perform on the given tasks?*
>
> The initial external knowledge policies barely succeeded in most of the tasks we evaluated. These results are plotted in Figure 3 with the label “BC”.
>
> 5. *The authors discussed broader impact and future work in Appendix D, but no limitations were addressed. Could the authors please add limitations to either Appendix D or the conclusion?*
>
> Yes! We have discussed the limitations of this work. Please refer to “General Response - Limitations”.

---

> > ### Comment · Reviewer_K32E · 2023-08-17
> > **Response to Author Rebuttal**
> >
> > Thank you for addressing my concerns! I can't see Figures 6 and 10 in the supplementary material nor the latest paper revision.

---

> > > ### Author Response · Authors · 2023-08-17
> > > **Thank you for letting us know!**
> > >
> > > Thank you for letting us know!
> > >
> > > There should be a link to the pdf with new experiments in the General Author Rebuttal. However, the link disappears when we check OpenReview today.
> > >
> > > We will let the program chairs and area chairs know about this.

---

> > > > ### Author Response · Authors · 2023-08-18
> > > > **The PDF with new experiments is up again!**
> > > >
> > > > The link to the PDF with new experiments is up again!
> > > >
> > > > You can find it at the bottom of the General Author Rebuttal.

---

> > > > > ### Comment · Reviewer_K32E · 2023-08-19
> > > > > **Response to Authors**
> > > > >
> > > > > Thank you for providing the link to the additional experiments. As such, I have increased my score.

---

### Official Review · Reviewer_CuzY · 2023-07-03

**Soundness:** 2 fair
**Presentation:** 3 good
**Contribution:** 3 good
**Rating:** 5
**Confidence:** 3

**Summary:**

The authors of this work introduce Knowledge-Grounded RL (KGRL), an RL framework that combines multiple knowledge policies to achieve human-like efficiency and flexibility in learning. They propose a new actor architecture called Knowledge-Inclusive Attention Network (KIAN) that enables arbitrary combinations and replacements of external policies through embedding-based attentive action prediction. KIAN also addresses entropy imbalance, a challenge in KGRL, addressing exploration in the environment.

**Strengths:**

Firstly, the paper addresses a relevant and interesting problem in the field of reinforcement learning (RL) by improving sample efficiency from arbitrary external policies and enabling knowledge transfer skills.
Furthermore, the paper is a well-written work that effectively conveys the ideas and findings in a clear and concise manner. The authors demonstrate excellent writing skills, employing appropriate terminology and organizing the content in a manner that enhances readability and comprehensibility.
In addition, the paper effectively establishes the motivation and position of the study in the existing literature. The authors articulate the significance and relevance of the research problem, demonstrating a strong understanding of the field. They situate their work within the broader scholarly context, highlighting how their study fills a gap in knowledge or builds upon prior research.
The methodology employed in the research is clearly described, allowing readers to understand and replicate the study. The authors provide a detailed and comprehensive explanation of the experimental and theoretical approach used, supported by well-designed figures and diagrams. These visual aids enhance the understanding of the methods employed and facilitate better comprehension of the research. Lastly, the authors provide solid mathematical understanding for their proposed method.

Collectively, these strengths highlight the quality of the paper. ts focus on addressing a relevant and interesting problem, combined with its well-written content, clear methodology, positioning in the literature, and mathematical rigor, make it an intersting contribution.

**Weaknesses:**

The paper has one notable weakness that should be addressed to enhance the overall quality. The evaluation presented in the paper is quite limited. The results obtained for the OpenAI robotics task only offer a slight support for the proposed method. The authors should consider expanding the evaluation to provide a more comprehensive assessment of the proposed method's performance.
Additionally, the use of only 3 or 5 seeds in the experiments may be insufficient, especially when considering the large error bars observed in some models. The authors should consider increasing the number of seeds to improve the statistical robustness of the results.
Moreover, the paper lacks clarity regarding the error bars shown in the plots. It is not explicitly stated what these error bars represent, which hinders the interpretation and understanding of the presented data.
Lastly, the paper would greatly benefit from conducting ablation studies to investigate the effects of various factors on the proposed method's performance. Specifically, the authors could consider performing ablation studies on
- the influence and distribution of attention/weights of actors,
- the impact of random/irrelevant policies in $\mathcal{G}$,
- the impact of the (near) optimal policy in $\mathcal{G}$,
- the impact of a larger set of knowledge policies $\mathcal{G}$,
- the effects of different types of knowledge policies,
- and an investigation of the mentioned entropy balance issue, that the authors specifically address in their method.
These ablation studies would provide valuable insights into the individual contributions and impacts of these factors on the overall approach.

Overall, addressing these weaknesses would significantly improve the research paper, clarifying important aspects such as error bars, conducting relevant ablation studies, ensuring consistency in reporting variances, and strengthening the empirical evidence for the proposed method.

### Minor
- I assume ori-KIAN is KIAN with the original $\mathcal{G}$, this should be mentioned in the text.
- l.314 mentions less variance, but tables do not show variances for individual experiments

**Questions:**

See the above mentioned ablation studies.

**Limitations:**

The limitations have not been discussed by the authors.
The above mentioned ablation studies will probably help to identify the shortcomings of the proposed method.

---

> ### Author Rebuttal · Authors · 2023-08-10
>
> 1. *The use of only 3 or 5 seeds in the experiments may be insufficient, especially when considering the large error bars observed in some models. The authors should consider increasing the number of seeds to improve the statistical robustness of the results.*
>
> Thank you for the suggestion! We have increased the number of random seeds run for all experiments. Please refer to “General Response - Experimental Significance”.
>
> 2. *The paper would greatly benefit from conducting ablation studies to investigate the effects of various factors on the proposed method's performance.*
>
> Thank you for the suggestion! We have conducted a series of ablation studies. Please refer to “General Response - Ablation Studies and More Analyses”.
>
> 3. *The paper lacks clarity regarding the error bars shown in the plots. It is not explicitly stated what these error bars represent, which hinders the interpretation and understanding of the presented data.*
>
> Thank you for pointing out the clarity issue! Each error band in all figures is a 95% confidence interval. In addition, we found that the colors for all algorithms in Figure 3 make it difficult to tell the error bands of KIAN, especially for OpenAI robotic tasks. Thus, we changed the colors of all curves in Figure 5. We will update the explanation of the error bands and figures in the final paper.
>
> 4. *The results obtained for the OpenAI robotics task only offer a slight support for the proposed method.*
>
> Actually, for Pick-and-Place and Push in Figure 3 (Figure 5) as well as Push->Pick-and-Place in Figure 4, KIAN converges much faster and has much smaller error bars compared to all other baselines. Also, in Tables 1 and 2 (zero-shot transfer experiments), KIAN’s performance is generally the best or very close to the best performance for OpenAI robotic tasks. Finally, in Appendix E, we showed that KIAN performs better in zero-shot transfer experiments with small variances for all OpenAI robotic tasks. We believe that these results suggest that KIAN is also effective for continuous-control tasks.
>
> 5. *I assume ori-KIAN is KIAN with the original , this should be mentioned in the text.*
>
> Thank you for pointing out this clarity issue! Yes, ori-KIAN is KIAN without using previously learned query and keys, which is the same setting as used in Figure 3 (Figure 5). We will clarify this in the final paper.
>
> 6. *l.314 mentions less variance, but tables do not show variances for individual experiments*
>
> Thank you for pointing out this clarity issue! We were limited by space when submitting the original paper, so we moved the tables with variances to Appendix E. We will clarify this in the final paper.
>
> 7. *The limitations have not been discussed by the authors. The above mentioned ablation studies will probably help to identify the shortcomings of the proposed method.*
>
> We have added limitations of our proposed method! Please refer to “General Response - Limitations”.
>
> 8. *The influence and distribution of attention/weights of actors:*
>
> Figure 6 in the additional-result document shows the weights of each knowledge policy after training. The results demonstrate different behaviors of KIAN in discrete decision-making and continuous control tasks.
>
> 8.a) Discrete decision-making: An agent learns to attend to different knowledge policies according to the current situation. Specifically, in a DoorKey-8x8 environment, the agent chooses to sequentially “pick up the key”, “open the door”, “search for the goal”, and “reach the goal”. Note that the attention to the inner policy does not disappear throughout the entire episode. This shows that the inner policy helps an agent explore the environment.
>
> 8.b) Continuous control: An agent learns to attend to only the inner knowledge policy. This result shows that KIAN enables effective knowledge transfer between external and inner policies. We believe this difference from the behaviors in discrete decision-making can be attributed to (1) how an action is sampled in continuous control tasks and (2) the suboptimality of external policies: Since an action is sampled by Gumbel softmax, only the action suggested by one knowledge policy will be applied, even though in probability, all knowledge policies are fused together. As a result, the agent may find it more efficient to let the inner policy first copy the useful skills from suboptimal external policies and focus on the inner policy to complete a task.

---

> ### Comment · Area_Chair_Leru · 2023-08-18
> **Reviewer Reponse Requested**
>
> Hello Reviewer,
>
> The authors have made efforts to address your comments on their work via the rebuttal. Part of the NeurIPS review process is participating meaningfully in the rebuttal phase to help ensure quality. Please read and respond to the author's comments today, latest tomorrow, to give everyone time to respond and reach proper conclusions.
>
> Thank you for your assistance in making NeurIPS a great conference for our community.

---

> > ### Comment · Reviewer_CuzY · 2023-08-21
> > **Answer to rebuttal**
> >
> > I thank the reviewers for providing the additional ablation studies, I have changed my score accordingly.

---

### Official Review · Reviewer_2LpZ · 2023-07-07

**Soundness:** 3 good
**Presentation:** 2 fair
**Contribution:** 3 good
**Rating:** 7
**Confidence:** 3

**Summary:**

Humans can learn by aggregating external knowledge from others’ policies of attempting a task. While prior studies in RL have incorporated external knowledge policies for sample efficiency,  there still remains the generalization problem to be solved that agents have difficulties to perform arbitrary combinations and replacements of external policies.
The authors propose a new actor architecture for Knowledge-Grounded RL (KGRL),  Knowledge-Inclusive Attention Network (KIAN), which allows free knowledge rearrangement due to embedding-based attentive action prediction.  KIAN addresses entropy imbalance as well. The authors demonstrate in experiments that KIAN outperforms other methods incorporating external knowledge policies under different environmental setups.

**Strengths:**

The authors clearly define the problem as how RL can be grounded on any given set of external knowledge policies to achieve knowledge-acquirable, sample-efficient, generalizable, compositional, and incremental properties.
The proposed method of KIAN is clearly described. They use knowledge keys and the query performs an attention operation to determine how an agent integrates all policies.
The solution of the entropy imbalance issues when integrating external policies are proposed as well.

**Weaknesses:**

Generally this work has many related works but is tackling the unique challenge problem of fusing knowledge policies with different state and action spaces.
Limitations of the proposed method is not clear based on the experimental results.

**Questions:**

More experiments directly connected with this challenging problem would more clearly support your claims.

**Limitations:**

Some insights and analysis about limitations based on the experimental environments more directly connected with the target problem are expected to be included.

---

> ### Author Rebuttal · Authors · 2023-08-10
>
> 1. *Generally this work has more related works than in the related work section, such as multi-task imitation learning etc,.*
>
> We understand that other works not listed in the related work section may seem to be addressing a similar problem as KGRL. However, we would like to point out that their central point is quite different from that of KGRL:
>
> a) What is our goal?
>
> Investigate (1) how existing policies can be fused to help an agent learn new tasks more efficiently and (2) how this learning process can be flexible, i.e., by freely reusing and updating the external knowledge-policy set without relearning how to navigate the entire policy set.
>
> b) What is not our goal?
>
> b.1) Hierarchical learning: We specifically discussed the differences between HRL and KGRL in Section 2. While HRL aims to decompose a complex task into a hierarchy of sub-tasks and learn a sub-policy for each sub-task, KGRL aims to address a task by observing and fusing a given set of external policies.
>
> b.2) Multi-task and continual learning: While multi-task RL aims to learn a set of closely related tasks concurrently [49], and continual RL aims to learn a sequence of tasks without forgetting previously learned skills [50,51], KGRL aims to learn a single task more efficiently and flexibly with the help of existing policies from any source while maintaining this efficiency and flexibility to a sequence of tasks.
>
> According to our goal, the works listed in Section 2 are better aligned with the scope of this work. However, we will add this discussion to the Appendix in the final paper!
>
> 2. *In addition to focusing on the fusion problem of given set of external policies, more explored approaches would be better.
> The formulation of the problem from broader view of use of external policies would be better.*
>
> Without a more concrete description of “explored approaches” and “broader view”, we find it difficult to respond to these comments. You seem to suggest a different scope for this work. However, we respectively disagree that a different scope would be better. We would like to restate our motivation for investigating KGRL: Unlike humans, current RL agents lack the ability to efficiently leverage external policies from different sources to help them learn new tasks. In addition, they lack the flexibility to learn with a changeable knowledge-policy set. Without this ability and flexibility, lots of resources will be wasted on (1) relearning the skills that are already developed by someone else and (2) re-exploring how different external policies may help an agent learn a given task. Therefore, we dedicate ourselves to studying KGRL, i.e., how an agent can be knowledge-acquirable, sample efficient, generalizable, compositional, and incremental.
>
> 3. *Some insights and analysis about limitations are expected to be included.*
>
> We have added insights/analyses and limitations of our proposed method! Please refer to “General Response - Ablation Studies and More Analyses” for insights/analyses and “General Response - Limitations” for limitations.
>
> **Reference:**
>
> [49] Vithayathil Varghese, Nelson, and Qusay H. Mahmoud. "A survey of multi-task deep reinforcement learning." Electronics 9.9 (2020): 1363.
>
> [50] Rolnick, David, et al. "Experience replay for continual learning." Advances in Neural Information Processing Systems 32 (2019).
>
> [51] Khetarpal, Khimya, et al. "Towards continual reinforcement learning: A review and perspectives." Journal of Artificial Intelligence Research 75 (2022): 1401-1476.

---

> ### Comment · Area_Chair_Leru · 2023-08-18
> **Reviewer Feedback requested**
>
> Hello Reviewer,
>
> The authors have made efforts to address your comments on their work via the rebuttal. Part of the NeurIPS review process is participating meaningfully in the rebuttal phase to help ensure quality. Please read and respond to the author's comments today, latest tomorrow, to give everyone time to respond and reach proper conclusions.
>
> Thank you for your assistance in making NeurIPS a great conference for our community.

---

> > ### Comment · Reviewer_2LpZ · 2023-08-21
> >
> > Thank you for the responses. I have read the reviews of the other reviewers and the authors' responses.  Also, I have read the related works and appendix again.
> > One of the misunderstanding of the paper comes from the misunderstanding of your problem of fusing knowledge policies with different state and action spaces that  enables efficient learning across a broader range of applications. My concerns is now solved.
> > Only other concern about limited evaluation has been also solved.
> > Finally I have changed and increased my score.

---

### Official Review · Reviewer_7i4w · 2023-07-07

**Soundness:** 3 good
**Presentation:** 3 good
**Contribution:** 3 good
**Rating:** 5
**Confidence:** 3

**Summary:**

This paper defines the Knowledge-Grounded RL setting, a general RL setting for integrating knowledge (in the form of policies) into a policy to learn new tasks efficiently. Essentially this setting is similar to the MDP setting except that the agent is also given a set of knowledge policies to utilize. The paper also introduces a system/architecture within this KGRL setting, called Knowledge-Inclusive Attention Network (KIAN). The aim is to improve RL that is grounded on external knowledge policies. The paper outlines five desirable human-like properties they desire in their agents: knowledge-acquirable, sample-efficient, generalizable, compositional, and incremental. Moreover, they formally define these so that they are measurable within the KGRL setting (e.g., for evaluating algorithms on these dimensions).

While previous methods typically intertwine knowledge representation and knowledge-fusion, thereby restricting their ability to adapt to numbers of policies, losing flexibility. KIAN is developed more flexibility, separating the knowledge representation and knowledge fusion.

KIAN consists of three components: a policy that learns a strategy (similar to a normal policy) called the internal, embeddings that represent the given knowledge (or external) policies, a query that performs attentive action prediction to fuse the internal and external policies.

KIAN also solves other issues that can occur in entropy-regularized KGRL. Entropy-regularized RL is common, but the authors show that in the KGRL setting issues can arise through entropy regularization where only a select few policies are selected, counterproductively reducing diversity in policy usage. The authors show that in the KGRL setting the agent will pay more attention to the policy with large entropy and in continuous control, will rely on the internal policy extensively. The paper introduces modifications so that this does not occur in KIAN.

The authors show results on both MiniGrid and robotics tasks and demonstrate sample efficiency, generalizability, as well as compositional and incremental learning.


**Strengths:**

The paper is mostly well-written and well-explained.

The method makes sense, and is a well-thought out architecture.

I like how the authors address the entropy imbalance problem.

I like that the authors define and quantify the behaviors they would like in the agent.

The results do seem to demonstrate their method is effective.


**Weaknesses:**

While I think the experiments are good, with appropriate baselines and good environments to test the agent’s capabilities. I am concerned about statistical significance. In particular, only 5 seeds are run, and the performance benefit in many cases is minimal, which may quite possibly be attributed to randomness. While I do believe the method outperforms the baselines, I cannot say so with a lot of confidence to merit inclusion at NeurIPS. If the authors can run more seeds, especially given the large variance, it would dramatically improve their results.


Qualms:
KGRL is consistently referred to as an RL framework, which it is, but the connotation can be misconstrued as being a framework “for” RL, implying it is a solution method for RL problems. I would recommend calling it a “setting” rather than a framework. Indeed, I was confused temporarily as a reader, especially when KGRL is stated as being “introduced” by this paper (as opposed to “described” or “outlined”).


Nits
Typo Line 345: “border” should be: “broader”


**Questions:**

In line 33, it is stated (for incremental): “Humans do not need to relearn how to navigate the entire knowledge set from scratch when they remove outdated strategies or add new ones”. Do humans truly “remove” outdated strategies?

In line 222, the paper states: “However, fusing multiple policies as equation (3) will make an agent biased toward a small set of knowledge policies when exploring the environment.” I am somewhat confused, equation (3) as is does not seem to have this issue. I thought this only occurs in the MaxEnt KGRL setting as introduced in the next section. Can the authors please clarify this?

The authors state the entropy imbalance problem is a property of the maximum entropy KGRL setting. I want to clarify that this is only shown for the specific case of policies in the form of equation (3), correct?

How could the policy embeddings/knowledge keys be learned?

Is Gymnasium Robotics used for the experiments? Or the older OpenAI codebase?


**Limitations:**

To me it is unclear how in more sophisticated settings how the knowledge keys (or policy embeddings) would be learned. This seems like a bottleneck to scalability that is not well-addressed.

---

> ### Author Rebuttal · Authors · 2023-08-10
>
> 1. *I am concerned about statistical significance. In particular, only 5 seeds are run, and the performance benefit in many cases is minimal, which may quite possibly be attributed to randomness.*
>
> Thank you for pointing this out! We have increased the number of random seeds run for all experiments. Please refer to “General Response - Experimental Significance”.
>
> 2. *Qualms: KGRL is consistently referred to as an RL framework, which it is, but the connotation can be misconstrued as being a framework “for” RL, implying it is a solution method for RL problems. I would recommend calling it a “setting” rather than a framework. Indeed, I was confused temporarily as a reader, especially when KGRL is stated as being “introduced” by this paper (as opposed to “described” or “outlined”).*
>
> Thank you for the suggestion! We agree that “setting” is a better term for this work and will update our final paper accordingly!
>
> 3. *Nits Typo Line 345: “border” should be: “broader”*
>
> Thank you for pointing out the typo! We will correct it in the paper :)
>
> 4. *In line 33, it is stated (for incremental): “Humans do not need to relearn how to navigate the entire knowledge set from scratch when they remove outdated strategies or add new ones”. Do humans truly “remove” outdated strategies?*
>
> We believe that when encountering a task, humans may collect a knowledge-policy set from a large pool of policies that they think might be useful for the task. If we find a policy in this smaller knowledge set that is no longer helpful for solving the task, we might remove it from the set.
>
> 5. *In line 222, the paper states: “However, fusing multiple policies as equation (3) will make an agent biased toward a small set of knowledge policies when exploring the environment.” I am somewhat confused, equation (3) as is does not seem to have this issue. I thought this only occurs in the MaxEnt KGRL setting as introduced in the next section. Can the authors please clarify this?
> The authors state the entropy imbalance problem is a property of the maximum entropy KGRL setting. I want to clarify that this is only shown for the specific case of policies in the form of equation (3), correct?*
>
> Yes, you are right! Fusing multiple policies with equation (3) will make an agent biased toward a small set of knowledge policies when exploring the environment by maximizing the entire policy’s entropy. We will clarify this point in the final paper.
>
> 6. *How could the policy embeddings/knowledge keys be learned?*
>
> They are jointly (end-to-end) learned with other components in KIAN. The gradient signals calculated by an RL algorithm, e.g., policy gradient, will be back-propagated to all policy embeddings. We provided the complete learning algorithm in Appendix A.6.
>
> 7. *Is Gymnasium Robotics used for the experiments? Or the older OpenAI codebase?*
>
> We use the older OpenAI codebase. The version we use lies within the Python package “gym” of version 0.21.0.
>
> 8. *To me it is unclear how in more sophisticated settings how the knowledge keys (or policy embeddings) would be learned. This seems like a bottleneck to scalability that is not well-addressed.*
>
> Each knowledge key is a trainable variable that can be end-to-end learned with other components of KIAN. Since each knowledge key is a point lying in an embedding space, adding more knowledge policies simply equals adding more points to the embedding space. As a result, KIAN can be easily scaled with a large group of external knowledge policies.

---

> > ### Comment · Reviewer_7i4w · 2023-08-14
> > **Initial Response to Authors**
> >
> > Thank you for answering all of my questions and making a faithful attempt to address my points.

---

> > > ### Comment · Reviewer_7i4w · 2023-08-16
> > > **Followup Response to the Authors**
> > >
> > > Thank you again for your response, and for addressing all my points. After reviewing the updated curves, I still maintain my score. For some environments, the results are statistically significantly better than the baselines. For most other environments, they are are not, though they do not appear to be worse. As such, I still believe the paper should be accepted. The evidence in support of the proposed method is convincing, but not overwhelming, hence why I am not giving a stronger accept.

---

### Author Rebuttal · Authors · 2023-08-10

We thank the reviewers for their insightful feedback! We address the shared reviewers’ comments below and will incorporate all feedback in the final paper.

### Experimental Significance
(Reviewer 7i4w and CuzY)

We have increased the number of random seeds to 10 for each experiment within the given time frame. Yet, we are still running with more random seeds. Also, we changed the colors of all curves in Figure 5 in the additional-result document for easier to read the error bands. Each error band in all figures is a 95% confidence interval. We will update the figures and explanation of the error bands in the final paper.

The results in Figure 5 show that even after adding more random seeds, KIAN converges much faster and reaches higher rewards for all tasks. In addition, KIAN’s error bands are small, indicating that its improved performance is statistically significant.

We observe similar results when running more random seeds for Figure 4. We apologize that due to time and resource limitations, we are not able to provide an updated plot for Figure 4 on time. Yet, we will update the results in the final paper.

### Ablation Studies and More Analyses
(Reviewer 2LpZ and CuzY)

* The impact of random/irrelevant policies in $\mathcal{G}$: Figure 7 in the additional-result document shows the learning curves of KIAN trained with 3 relevant, 6 (3 relevant + 3 irrelevant), and 9 (3 relevant + 6 irrelevant) external policies for MiniGrid Unlock. The results demonstrate that adding redundant/irrelevant knowledge policies slightly slows down the convergence speed but still reaches high rewards with small error bands. This slight convergence-speed decrease is quite as expected since the agent needs to (1) learn to ignore the policies that are not helpful for the task and (2) navigate through a larger set of external knowledge policies.

* The impact of the (near) optimal policy in $\mathcal{G}$: Figure 8 in the additional-result document shows the results of running FetchReach-v1 with an optimal external knowledge policy. We did this ablation study on FetchReach-v1 since we can only obtain the optimal policy for this environment without doing further training.

    The results demonstrate that with optimal external policies, SAC-RL, SAC-KoGuN, and SAC-KIAN achieve similar performance, but SAC-KIAN still performs slightly better than others.

* The impact of a larger set of knowledge policies $\mathcal{G}$: In Figure 7 in the additional-result document, we can also see the effects of learning with a larger set of external policies. The results show that the convergence speed slows down when the knowledge set is expanded with more policies that are irrelevant to the task.

    We believe that one of the reasons for slower convergence speed is that the agent needs to navigate through more policies before it knows which ones are useful for the task. Therefore, when the knowledge-policy set is really large, KGRL methods, including KoGuN, A2T, and KIAN, are not guaranteed to be more efficient than RL methods. We will also discuss this point in the limitations.

* The effects of different types of knowledge policies: Actually, KIAN’s results in Figure 4 (in the original paper) are run with an external knowledge set that includes two different types of knowledge policies: if-else-based programs and neural networks. The results demonstrate that mixing different types of knowledge policies will not hurt performance at all. This observation is as we expected since KIAN only cares about the state-action mappings provided by each knowledge policy instead of how this mapping is generated.

* An investigation of the mentioned entropy balance issue: Figure 9 in the additional-result document shows the results of KIAN with and without the entropy imbalance issue addressed. The results show that without addressing the issue with equation (9), KIAN cannot fully benefit from the guidance provided by external policies. In addition, KIAN may perform even worse than RL with no external knowledge policies (see the results of FetchPickAndPlace and FetchPush). The reason is provided by Proposition 4.3.

    These results demonstrate that (1) entropy imbalance is indeed an issue for KGRL and (2) our proposed method can largely mitigate this issue.



### Limitations
(Reviewer 2LpZ, CuzY, K32E)

Here are the limitations we will summarize in Conclusion/Appendix D:
* If the external knowledge set is very large and contains irrelevant policies, the efficiency of KIAN, and other existing KGRL methods, may decrease. This is discussed in “General Response - 2.4”.
* The current design of KIAN requires external and inner policies to share the same state and action space. Therefore, if a policy for Task A includes a skill that can benefit Task B, but Task A and B have different state and action space, the policy cannot be used as an external policy for Task B. This can limit the number of existing policies that can be used to help learn a new task.

---

> ### Comment · Reviewer_7i4w · 2023-08-14
> **Followup Question for Authors**
>
> What did the error bars in the original pdf represent?

---

> > ### Author Response · Authors · 2023-08-14
> > **Error Bars in the Original PDF**
> >
> > Thank you for the question!
> >
> > The error bars in the original pdf are also 95% confidence interval.

---

### Comment · Area_Chair_Leru · 2023-08-16
**Reviewer Responses Needed**

Hello Reviewers,

The authors have made efforts to address your comments on their work via the rebuttal. Part of the NeurIPS review process is participating meaningfully in the rebuttal phase to help ensure quality. Please read and respond to the author's comments today, latest tomorrow, to give everyone time to respond and reach proper conclusions.

Thank you for your assistance in making NeurIPS a great conference for our community.

---

### Decision · Program_Chairs · 2023-09-21

**Decision:**

Accept (poster)

**Comment:**

This paper proposes a Knowledge-Grounded RL for integrating knowledge (in the form of policies) into a policy to learn new tasks efficiently. The paper outlines five desirable human-like properties they desire in their agents: knowledge-acquirable, sample-efficient, generalizable, compositional, and incremental. The authors have shown the value of this work via additional experiments, and the reviewers have not found technical flaws in the work. Most reviewers appreciate how the work studies the entropy imbalance problem in entropy-regularized KGRL.